# Impacts of hydrofacies geometry designed from seismic refraction tomography on estimated hydrogeophysical variables

Nolwenn Lesparre[1], Sylvain Pasquet[2,*,**], Philippe Ackerer[1]

[1]Université de Strasbourg, CNRS, EOST, ENGEES, Institut Terre et Environnement de Strasbourg, UMR 7063, 5 rue Descartes, Strasbourg F-67084, France

[2] Université Paris Cité, Institut de physique du globe de Paris, CNRS, Paris, France

* now at : UAR 3455 OSU ECCE TERRA, CNRS, Sorbonne Université, 75005 Paris, France and ** UMR 7619 METIS, CNRS, Sorbonne Université, EPHE, 75005 Paris, France

*Correspondence to*: Nolwenn Lesparre (lesparre@unistra.fr), Sylvain Pasquet (sylvain.pasquet@sorbonne-universite.fr), Philippe Ackerer (ackerer@unistra.fr)

**Abstract.** Understanding the critical zone processes related to groundwater flows relies on subsurface structure knowledge and its associated parameters. We propose a methodology to draw the patterns of the subsurface critical zone at the catchment scale from seismic refraction data and show its interest for hydrological modelling. The designed patterns define the structure for a physically based distributed hydrological model applied to a mountainous catchment. In that goal, we acquired 10 seismic profiles covering the different geomorphology zones of the studied catchment. We develop a methodology to analyze the geostatistical characteristics of the seismic data and interpolate them over the whole catchment. The applied geostatistical model considers the scale variability of the subsurface structures observed from the seismic data analysis. We use compressional seismic wave velocity thresholds to identify the depth of the soil and saprolite bottom boundaries. Assuming that such porous compartments host the main part of the active aquifer, their patterns are embedded in a distributed hydrological model. We examine the sensitivity of classical hydrological data (piezometric heads) and geophysical data (magnetic resonance soundings) to the applied velocity thresholds used to define the soil and saprolite boundaries. Different sets of hydrogeological parameters are used in order to distinguish general trends or specificities related to the choice of the parameter values. The application of the methodology to an actual catchment illustrates the interest of seismic refraction to constrain the structure of the critical zone subsurface compartments. The sensitivity tests highlight the complementarity of the analyzed hydrogeophysical data sets.

# 1 Introduction

Groundwater flow and catchment discharge are strongly controlled by the structure of the critical zone (CZ) underground part and its related hydraulic properties and boundary conditions (Cassidy et al., 2014; Diek et al., 2014; Fleckenstein et al., 2006; Gabrielli et al., 2012). The bottom limit of the CZ corresponds to the base of the aquifer above which alteration of underground materials typically increases towards the surface (Anderson et al., 2007; Brantley et al., 2007). From the substratum to the soil surface, the deeper weathered rocks progressively evolve to saprolite and soil, designing the compartments of the underground CZ classical scheme (Anderson et al., 2007). The porosity and hydraulic conductivity of such compartments increase toward the surface as fractures, weathering, and alteration processes enlarge the porous space and ease water flows (Brooks et al., 2015). In the following, we refer to the soil as the superficial compartment made of disaggregated materials, while the saprolite denotes deeper weathered materials with a lower porosity. In mountainous environments, the subsurface is particularly heterogeneous, and the weathered bedrock, saprolite, and soil compartments can show significant thickness variations at the catchment scale (Befus et al., 2011; Diek et al., 2014; Koch et al., 2009; St. Clair et al., 2015). The thickness variability of these CZ underground compartments is related to the history of climate, weathering and erosion processes, regional tectonic forcing, or the occurrence of some metamorphic intrusions. These processes might have different impacts across the catchment due to local topography and lithology (Anderson et al., 2007; Holbrook et al., 2019; Rempe and Dietrich, 2014; Riebe et al., 2017).

The spatial distribution of the subsurface hydraulic properties determines the way infiltrated water drains into storage areas (Brooks et al., 2015). It has been shown that the thickness distribution of the underground CZ compartments impacts the watershed's water budget (Bertoldi and Rigon, 2006; Lanni et al., 2012). Moreover, the hydrogeological facies geometry is a key property for understanding the dynamic of the groundwater from piezometric measurements (piezometers may intercept different layers). Knowing the hydrogeological facies geometry is thus crucial to design hydrogeological models, especially under phreatic conditions (Carrera et al., 2005). However, the inverse problems that seek the hydraulic parameters of distributed hydrologic models applied at the catchment scale are known to show a strong non-uniqueness (Ebel and Loague, 2006). The value of the hydraulic parameters related to each mesh element of such models has to be determined, while the available measurements might be equally fitted by different sets of properties. To tackle this non-uniqueness issue, spatialized observations providing information on diverse hydraulic properties should be integrated in the inversion process (Zhou et al. 2014). Nevertheless, hidden by nature, the measurement of water storage and flow properties in the underground complex structure is still arduous. Basic but crucial information, such as the interface geometry between the different CZ underground compartments, are often missing (Brooks et al., 2015). Recent studies show that the characterization of the underground CZ structure remains challenging (Flinchum et al., 2018; Gourdol et al., 2020; Kaufmann et al., 2020, Pasquet et al., 2022).

Geophysical imaging methods provide an insight into the underground CZ architecture as they furnish a vision of the subsurface geophysical properties with a continuous spatial coverage along acquisition profiles. In particular, seismic refraction tomography (SRT) supplies structural information of the CZ underground part. SRT highlights the spatial variability

of the subsurface properties and can be used to distinguish characteristic patterns (Befus et al., 2011; Cassidy et al., 2014; Dal Bo et al., 2019; Olona et al., 2010; Huang et al., 2021). The inversion of SRT data provides a distribution of compressional P waves velocity ($v_p$) in the subsurface, which depends mainly on the medium's porosity, density and mineralogy. Weathering processes occurring in the CZ induce a decrease of $v_p$ by increasing the degree of fracturation and porosity. Indeed, $v_p$ is slower in pores filled by air or water than in the rock matrix (Pasquet et al., 2015; Parsekian et al., 2015). Moreover, $v_p$ is lower in secondary minerals (i.e., clays, oxides) than in parent minerals (i.e., quartz, plagioclase) (Olona et al., 2010; Parsekian et al., 2015). SRT is thus well suited to distinguish the spatial variability of the interfaces between the CZ underground compartments (Befus et al., 2011; Flinchum et al., 2018; Holbrook et al., 2013; Olona et al., 2010; Olyphant et al., 2016; Huang et al., 2021, Pasquet et al., 2022; St. Clair et al., 2015).

In this study, we assume that the subsurface description obtained by SRT holds also for hydraulic properties, i.e. the layers distinguished with SRT present a homogeneous porosity or hydraulic conductivity at the catchment scale. From this assumption, we assess the impact of the underground geometry based on SRT on variables dependent on the groundwater storage estimated from hydrological modelling. We use parameter values obtained from previous studies in the Strengbach catchment (Belfort et al., 2018; Lesparre et al., 2020) to perform this analysis under field site conditions. The following methodology is applied:

1.      Measured SRT profiles are analysed in a geostatistical framework using filtering, truncated power value variograms due to the change in measurement scale with depth, and 250 three dimensional (3D) velocity fields are generated over the catchment;

2.      A threshold velocity is prescribed to estimate the layers' thicknesses for the 250 velocity fields;

3.      Numerical simulations are performed for the 250 geometries using a hydrological physically-based model of the water catchment and uniform hydraulic parameters for each layer. Model outputs of interest are piezometric heads, i.e. levels of the saturated zone as the groundwater is unconfined and Magnetic Resonance Soundings (MRS), which include information in the water content of the unsaturated zone.

4.      The impact of the geometry and additional uncertainties on threshold velocity values and hydraulic parameters is evaluated by a simplified sensitivity analysis of MRS and piezometric heads. We focused the analysis on spatialized data as they better show how the data sensitivity to the tested conditions depends on the local context (i.e. steep slopes leading to drainage or flat regions favouring storage).

The originality of this paper lies in the framework developed here to build the hydrological model geometry from SRT data and assess how variables informing on the groundwater state estimated from that model are sensitive to the geometry uncertainty. The field site and the treatment of the SRT data are described in section 2 and 3, respectively. The construction of the 3D geometries from the SRT data and the geostatistical analysis are explained and discussed in section 4. Then, the

hydrological model NIHM and its related outputs are exposed in section 5. Finally, we evaluate how the simulated water circulations are impacted by the model interface geometry through an analysis of measurable data sensitivity in section 6.

## 2 Field context

### 2.1 Studied site

The Strengbach watershed is located in the Vosges Mountains (Northeast France) and covers an area of about 0.8 km² (Fig. 1).

The elevation ranges between 883 and 1146 m, and the topography is rugged with incised slopes that can reach up to 30°. The catchment is divided into two hillsides with different morphology and meteorology influenced by their respective orientation. The southeast slopes are gentler; the temperature is usually lower and associated with higher precipitation than the northwestern slopes.

The subsurface can be described using the classical CZ scheme with a degree of weathering and fracturation that increases

from depth towards the surface (Brantley et al., 2017). Most of the catchment lies on a Hercynian granitic bedrock, but micro-granite and gneiss constitute the protolith of the southern and northern crests, respectively (El Gh'Mari, 1995). That hard-rock level may be locally fractured and is overlaid by weathered bedrock made of chemically altered and fractured rocks. When sufficiently altered, this weathered bedrock turns into saprolite, forming a sandy coarse-grain matrix containing gravels and pebbles (Fichter et al., 1998a). Then, the soil composes the uppermost layer. The physical properties of each of these two

layers and their respective thicknesses vary spatially over the catchment, as expected by El Gh'Mari (1995) and confirmed by a recent hydrogeophysical study (Lesparre et al., 2020a). Some catchment regions, such as crests, slopes and valley bottom might present distinct soil and saprolite thicknesses and varying porosity distributions. The exposure and the inclination could also impact thickness and porosity distributions, allowing observations of differences from one hillside to another. The weathering history of the hillsides also differs as hydrothermal circulations altered the northern slope 180 My ago (Fichter et

al., 1998b).

### 2.2 Meteorological and hydrological observations

The Strengbach catchment is a well-studied research site that hosts numerous scientific investigations spanning various key questions concerning the functioning and the vulnerability of the CZ (Pierret et al., 2018). Permanent measurement stations have continuously monitored the meteorological and environmental conditions since 1986. These long-term observations are

managed by the Observatoire Hydro-Géochimique de l'Environnement (OHGE, http://ohge.unistra.fr; CNRS/University of Strasbourg), which is part of the French network of CZ observatories (OZCAR; Gaillardet et al., 2018). OHGE also provides convenient facilities for punctual scientific experiments (Pierret et al., 2018).

The meteorological forcing is monitored at two stations: one placed on the northern crest of the catchment and the other settled near the outlet (Fig. 1). Both stations record rainfall, temperature and relative humidity. The upper station also monitors global

radiation, wind speed and snow thickness. Seven rain gauges provide regular measurements covering the catchment to infer rainfall spatial variability.

The stream flow rate is continuously monitored at the catchment outlet with an H-flume (RS station). A second flume records the flow rate upstream (RAZS station). The underground structure of the catchment was investigated by drilling nine boreholes with depths from 15 to 120 m. Three boreholes were cored to provide direct insight into the deep CZ structure. Most of the

boreholes intercept fractures in the bedrock, such that monitored water levels do not necessarily display hydraulic pressure heads corresponding to the water flow dynamics in the shallow porous medium. Recently, 10 piezometers were drilled to depths lower than 7 m and have recorded data since September 2020. Those piezometers are spatially distributed over the catchment with a few of them installed near the original boreholes.

### 2.3 Hydrogeological knowledge

A combined analysis of the catchment pedology and MRS data covering the catchment has shown a relatively flat region of water storage upstream the creek main spring (Boucher et al., 2015; Lesparre et al., 2020a). A previous analysis of MRS measurements rendered a qualitative depiction of the subsurface water volume distribution in the catchment (Boucher et al., 2015; Pierret et al., 2018). The map of the water volume concealed in the weathered layer shows significant variability strongly correlated with the pedologic zonation. Low water contents are suggested on the northern crest by MRS measurements, but

clayey materials covering that region might prohibit the detection of subsurface water (Boucher et al., 2015). On the northern hillside, the shallow subsurface is made of fissured/fractured granites that intense hydrothermal circulations have altered in the past. There, the estimated water volume is intermediate and seems to feed perennial flow over time (Boucher et al., 2015). The southern hillside, which appears to be less weathered (Fichter et al., 1998b), shows lower water content with a drier vadose zone less prone to infiltration or better drained than the northern hillside.

Finally, water content is higher underneath the wetland in the downstream part of the catchment and under the flat colluvium zone, which most likely corresponds to the thickest porous subarea of the catchment (Boucher et al., 2015). MRS signals recorded in that zone (called zone 2 in Lesparre et al., 2020a, see Fig. 1) show higher amplitudes than other acquisition locations, indicating a higher water content at depth. This can be explained by a thicker water bearing unit in that zone, as suggested by the results of the hydrological modeling (Lesparre et al., 2020a).

**3 Seismic refraction data**

### 3.1     Seismic velocity profiles

Ten SRT profiles, covering a total length of 2 km, were acquired in June 2018 and August 2019. Their locations were chosen to cover specific areas of the catchment, such as the valley bottom, the crests, the region upstream of the creek spring and both hillsides (Fig. 1). The surveys were designed to explore how the underground part of the CZ evolves in these different regions,

which were previously distinguished by a joint analysis of pedological and MRS data collected across the catchment (Boucher et al., 2015; Lesparre et al., 2020a). Seismic data were collected using up to 6 24-channel seismic recorders (Geometrics) and 14-Hz vertical-component geophones spaced with 2 m. For each profile, we used either 72, 96 or 144 geophones, for total lengths up to 142 m, 190 m and 286 m, respectively (Table 1). The source signal was generated with 4 stacks of a 5 kg sledgehammer blow on a metal plate, with shots located every other 5 or 6 geophones, starting at first and ending at last

geophone.

First arrival times were picked manually on each shot gather. Signal-to-noise ratio varies significantly for each profile, but is mostly high enough to confidently identify first breaks up to 100-150 m distance from the source (Fig. 2). This is more than enough to characterize the granite weathered zone anticipated to extend down to 10-15 m at most in such mountainous temperate catchment. The observed travel times were associated with a 5% picking error, then used to build the subsurface P-

wave velocity structure ($v_p$) by solving an inverse problem with the pyGIMLi refraction tomography inversion module (Rücker et al., 2017). In pyGIMLi, the inversion domain corresponds to a triangular mesh with cells of constant velocity through which rays are traced using a shortest-path algorithm (Dijkstra, 1959; Moser, 1991). The velocity in each mesh cell is estimated using a generalized Gauss-Newton inversion framework. The inversion is iterative and starts with an initial model consisting of a velocity field that increases linearly with depth from [250 - 750] m/s at surface to [2000 – 5000] m/s in depth

(Table S1). The velocity field is then smoothly updated at each iteration in order to reach the closest match between predicted and observed travel times. Inversions were performed with 144 combinations of starting models and regularization parameters (Table S1) in order to explore the possible solutions and estimate the uncertainty of the velocity distribution along each profile (Pasquet et al., 2016). A selection is then applied to keep only the results of inversions performed with a set of parameters that obtained a root mean square error < 2.5 ms and a root mean square error weighted by the variance $\chi^2 < 2$, for all 10 profiles

where $\chi^2 = \dfrac{(d_{obs} - d_{est})^2}{\varepsilon_{obs}^2}$, with $d_{obs}$ and $d_{est}$ the measured and estimated travel times, respectively and $\varepsilon_{obs}$ the travel time measurement error. We applied a systematic error of 5% on each picked travel time, setting $\varepsilon_{obs}$ lower and upper bounds at 0.3 and 3 ms, respectively. Among the 144 combinations of starting models and regularization parameters, 104 inversion results fulfill these requirements for all 10 profiles. The mean and the standard deviation of $v_p$ are then computed for each pixel of the SRT profiles from the 104 selected models (Fig. A1 and S1). The standard deviation distribution provides an

estimate of the $v_p$ variations' likelihood.

Each seismic profile inversion result is extracted to build horizontal maps of the average $v_p$ distributions at different depths (Fig. 3). Each profile was flattened, so the depth of each point corresponds to its orthogonal distance to the surface. The standard representations of the seismic profiles (distance vs. elevation) are also given in Fig. A1. $v_p$ varies globally between

400 m/s and 4500 m/s and increases progressively downward. Above a depth of 3 m, $v_p$ values are globally homogeneous and remain below 700 m/s (Fig. 3). At a depth of 3 m, profiles are heterogeneous with $v_p$ varying in between 700 and 2000 m/s. At depths of 5 and 8 m, profile 1 and large parts of profiles 2 and 3 show low $v_p$ values with a discrepancy of 1000 m/s compared to the other profiles. At a depth of 24 m, $v_p$ is again homogeneous with values above 2700 m/s for all profiles.

## 3.2     Underground structure along the SRT profiles

We explain the progressive increase of $v_p$ downward by a decreasing of weathered materials with depth, as observed in other sites lying on crystalline or rhyolitic bedrocks (Befus et al., 2011; Holbrook et al., 2014; Olyphant et al., 2016). $v_p$ variations observed from a profile to another at a 5 m depth suggest that the thickness of the weathered medium varies in different areas of the catchment. Results obtained along profile 1 show that the region upstream the main spring presents a thicker weathered zone compared to the rest of the catchment. The same conclusion was previously deduced from MRS measurements showing a region with higher water content (Boucher et al., 2015). In Lesparre et al. (2020a), MRS data estimated by the hydrological model NIHM (described below) were fitted to field measurements in order to calibrate the thickness and the porosity of the model. This calibration showed that a thicker weathered zone was required in that same area upstream of the main spring. The SRT data confirm the occurrence of that deeper weathered zone that is not limited around the MRS acquisition station but extends all along SRT profile 1. Our results also reveal that a thicker weathered region is susceptible to occur at the bottom of steep slopes as shown in profiles 2 and 3 (Fig. A1). Alternatively, weathered materials located in the valley bottom may be relatively thinner than other regions. Discrepancies are noticed from one slope to another, notably along the third profile, but no particular trend can be extracted to distinguish the north- and south-facing slopes.

In the Strengbach catchment, the underground porous material is described by two layers: the soil and the saprolite (El Gh'Mari 1995; Fichter et al., 1998a; Lesparre et al., 2020a). From the literature, only a few studies have explored the choice of a velocity threshold to delimit the saprolite upper and lower interfaces in such hard-rock contexts. Begonha and Sequeira Braga (2002) measured ultrasonic velocities on saprolite and weathered granite samples from Oporto (Portugal). They showed that porosity is the most influential property on the seismic velocity when studying the influence of weathering. Their analysis of 167 samples concluded that the velocity threshold between saprolite and moderately weathered granite is around 2000 m/s. Several field SRT measurements above crystalline bedrocks have confirmed this threshold value by comparing the profiles with pits, borehole logs or images acquired with other geophysical methods (Olona et al., 2010; Befus et al., 2011; Holbrook et al. 2014).

Other studies allocated the saprolite bottom interface at the depth where $v_p$ exceeds either 1100 m/s, 1200 m/s or 1400 m/s

(Flinchum et al., 2018; Holbrook et al., 2019). The range of $v_p$ in soil is less discussed because SRT is not always efficient in providing information with a fine-enough resolution to study such a thin layer. The resolution depends on the inter-distance between geophones, and for studies exploring the protolith upper interface, long inter-distances between geophones are preferred. Moreover, ultrasonic measurements on soil samples raise issues concerning preserving the in-situ conditions of the medium analyzed. In a similar crystalline context, Befus et al. (2011) performed SRT using a 1-m spacing between geophones to delimit soil < 0.5 m thick. They estimated that $v_p < 700 \text{ m/s}$ corresponded to the interface between these disaggregated materials and saprolite.

On the Strengbach catchment, different boreholes and pits were excavated to study the soil properties, the structure of the shallow underground CZ, and the erosion processes (Ackerer et al., 2016; Belfort et al., 2018). Unfortunately, the pits are distant by more than 100 m from the SRT profiles. We had to consider the steep slopes and the density of the vegetation when designing the layout of SRT surveys. Thus, we initiate our analysis by only examining $v_p$ variations along the profiles that are distant by less than 50 m from a borehole to provide an order of magnitude of the $v_p$ thresholds at the soil and saprolite interfaces. In that goal, we consider $v_p$ values corresponding to the interfaces depth of the soil and saprolite identified when drilling the boreholes (Table 2). The soil thickness is not precisely estimated from the boreholes drilling as it is generally thin at the drilling locations (i.e., around 0.5 m and never above 1 m thick). The $v_p$ threshold of the soil bottom interface varies in [410; 720] m/s along profiles 9, 13 and 3, which are distant by less than 35 m from the F1, Pz3 and Pz10b boreholes, respectively (Table 2). The saprolite bottom interface is estimated as the depth where the drilling tool had to be changed as it was penetrating a much less weathered rock. This interface is estimated at a depth of 4.5 m in the Pz3 borehole located close to profile 13 (20 m). For that borehole, the $v_p$ saprolite threshold varies in [1480; 2245] m/s (Table 2). The correspondence between the F1 borehole and its closest SRT profile gives a much lower velocity range in [900; 1030] m/s. This lower range can be explained by the comparison between a local measurement of the saprolite bottom location from the drilling, while on the SRT profiles the resolution is of a few meters so local heterogeneities are smoothed. All the more, there is more distance (35 m) between the F1 borehole and its nearest profile compared to the other boreholes and their respective neighboring profiles. The F8 borehole that is close to the 3rd profile is excluded from our analysis since the borehole is located in the valley bottom where the SRT profile shows a strong heterogeneity and, therefore, a much wider velocity range (Fig. A1 and Table 2). To assess the variability of the soil (saprolite) compartment thickness along each profile, we apply a $v_p$ threshold of 700 m/s (2000 m/s) (Fig. S2). We note that the average thickness of 3 m soil in zone 3 is twice as high as in the other zones (Fig. S3). The average thickness estimated for the saprolite is around 3.5 m in zones 1 and 4, while it reaches 8 m in zone 3 and 12 m in zone 2 (Fig. S3).

# 4  Construction of 3D $v_p$ models from SRT

## 4.1  SRT data filtering

Geostatistical tools are applied to interpolate $v_p$ in order to construct 3D $v_p$ blocks that could help in defining the geometry of the hydrological model covering the whole catchment. As mentioned above, velocity trends are observed with depth due to weathering processes related to changes in porous material properties along the profiles (Fig. 2). Besides, at a given depth we observe strong $v_p$ variations at the catchment scale leading to strong variations of the soil and saprolite thicknesses obtained with a fixed $v_p$ threshold (Fig. S2). Since $v_p$ maps show non stationary significant variations, SRT data have to be filtered to remove these trends and perform the geostatistical analyses. In that goal, the water catchment is partitioned in zones. The zonation and the filtering of the $v_p$ values are performed in four steps:

1.      We analyze the SRT profiles that show strong lateral variations to distinguish eventual locations of zones' boundaries crossing SRT profiles.

2.      Since the SRT profiles do not cover the whole catchment, we add constraints on zones delimitation by considering the soil surface slope (Fig. 4a) and altitude (Fig. 4b). We chose these two variables because we assume that the evolution of the porous material is linked to erosion and weathering processes, which both depend on slope and altitude (Riebe et al., 2017). The validity of a such hypothesis is confirmed by Uhlemann et al. (2022). Slope and altitude thresholds are defined from the analysis of a digital elevation model (DEM) characterized by a 0.5 m lateral resolution. The slopes are computed after applying a $40 \times 40$ m rectangular filter to remove the effects of the small-scale asperities of the topography. The thresholds are determined so the zonation is consistent with the lateral variations observed on the seismic profiles (Fig. 4c). We favor a limited number of four zones for having enough data in each zone to compute reliable statistics.

3.      For each zone $i$, an average velocity at a given depth $<v_p^i(z)>$ is computed (Fig. 5). Close to the surface (depth < 2 m), $v_p$ distributions are similar from one zone to another (Fig. 5 and S4). Deeper, $v_p$ increases faster in zones 1 and 4, with a similar behavior until $v_p > 2000$ m/s, which corresponds to a depth of about 7 m. In the remaining two zones, $v_p$ increases faster in zone 2 down to a depth of 7 m, where $v_p$ starts to increase faster in zone 3 instead.

4.      Each SRT profile is split according to the zonation (Fig. 4c) and for each sub-profile corresponding to zone $i$, the residual is computed using $w = \log 10\left(v_p\right) - <\log 10\left(v_p^i(z)\right)>$. The logarithm of the velocity is used because its distribution is closer to a Gaussian distribution than the velocity itself. $<\log 10\left(v_p^i(z)\right)>$ represents the average log-velocity at a depth $z$ in zone $i$.

The result of the procedure is presented in Fig. 6a and 6b for profile 2. The trend with depth and the contrast in velocity at the interface between two zones at a distance of 170 m can be seen in figure 6a. After filtering, the residuals do not show any

vertical trend but some minor differences still remain at the interface between zones 3 and 4 (blue clear and yellow lines above the profiles Fig. 6b).

## 4.2 Geostatistical modelling of the seismic P velocities

In preliminary tests, horizontal and vertical variograms were estimated without considering the zonation of the catchment. In the XY plane, the variogram shows a horizontal coherency (blue line, Fig. S5), but no vertical correlation arises (red line,

Fig. S5). Therefore, variograms for horizontal slices of 0.5 m are computed from the surface down to a depth of 25 m.

We chose the truncated power value (TPV) model to fit each experimental variogram because the support volume of SRT measurements increases with distance between two geophones (Di Federico and Neuman, 1997; Neuman et al., 2008). The

TPV model filters out random fields with an integral scale larger than $\lambda_u$ and lower than $\lambda_l$ (Di Federico and Neuman 1997;

Neuman et al., 2008). $\lambda_u$ is assimilated to the dimension of the sampling scale — here the catchment size — while $\lambda_l$ refers

to the data support— in our case the SRT resolution (Heße et al., 2014; Neuman et al., 2008). The TPV variogram $\gamma(s, n_l, n_u)$ is defined as:

$$\gamma(s, n_l, n_u) = c_0 + \gamma(s, n_l) - \gamma(s, n_u),$$

(1)

with $s$ the lag distance, $\gamma(s, n_l)$ the variogram associated with the lower wave number $n_l = 1/\lambda_u$ and $\gamma(s, n_u)$ the

variogram related to the upper wave number $n_u = 1/\lambda_l$. $c_0$ corresponds to the nugget and is directly determined by the

variance of the 104 seismic results obtained for each profile with $s = 0$. TPV models can be characterized either by a Gaussian or an exponential variogram. In our case the Gaussian TPV variogram better fits the experimental variogram and writes:

$$\gamma(s, n_m) = \sigma(n_m)^2 [1 - \exp(-\frac{\pi}{4}(sn_m)^2 + (\frac{\pi}{4}(sn_m)^2)^H \Gamma(1 - H, \frac{\pi}{4}(sn_m)^2)],$$

(2)

$$\sigma(n_m)^2 = \frac{C}{2Hn_m^{2H}}$$

where $\Gamma$ represents the gamma function, the variance $\sigma(n_m)^2 = \frac{C}{2Hn_m^{2H}}$, $0 < H < 1$ is the Hurst coefficient (Hurst,

1951) and C is a constant. $m$ represents either the index $u$ or $l$, of the upper or lower wave number, respectively.

One theoretical variogram is estimated in each 0.5 m horizontal layer and we analyze the evolution of each variogram

characteristics with depth. In that goal, we compute the values of $s_p$ and $\gamma(s_p)$ that correspond respectively to the abscissa

and ordinate of the point where the variograms reach a plateau (yellow stars, Fig. 7). $\gamma(s_p)$ is estimated as the theoretical

variogram average when $s > 200\,\text{m}$ and is associated to the variance of the variogram. $s_p$ corresponds to the projected lag distance where the theoretical variogram reaches $\gamma(s_p)$ and is related to the correlation length of the TPV variogram. $s_p$ is

constant until a depth of 7.5, where the variable jumps abruptly before decaying progressively (Fig. 8a). $\gamma(s_p)$ increases to a depth of 3 m, then it decreases with depth (Fig. 8b). The nugget, $c_0$, shows a strong decrease between the surface and a depth of 1 m, where it stabilizes until a depth 5 m before it increases with depth (Fig. 8c).

$s_p$, $\gamma(s_p)$ and $c_0$ variations are influenced by the acquisition geometry of the SRT data. Since the sensors are installed on the surface, the resolution is more accurate in the shallow medium in between 2 and 6 m depth. Smaller targets can be detected

near surface so smaller $s_p$ values are observed. This better accuracy is confirmed by the lowest $c_0$ values, and the largest $\gamma(s_p)$ reflecting the medium heterogeneity. The regularization process used during the SRT inversion involves smoothing the $v_p$ distribution. The less-constrained deeper region is depicted by more laterally extended (higher $s_p$ values) and blurred targets (lower $\gamma(s_p)$). The limited resolution of SRT in the very shallow media explains the low $\gamma(s_p)$ values in the medium close to the surface. It is impossible to resolve targets with a smaller size than the distance between the geophones. The depth

of 3 m at which $\gamma(s_p)$ is maximum is similar to the geophones inter-distance (i.e., 2 m). This explains as well the higher values of $c_0$ in the first meter below surface compared to the underlying region.

Beyond the acquisition geometry and the characteristics of SRT images related to the inversion process, $s_p$ and $\gamma(s_p)$ variations with depth can be explained by the structure of the underground medium. The $s_p$ abrupt jump could be related to the transition where the medium becomes more coherent. In the shallow region, the strongly weathered medium is composed

of materials presenting smaller characteristic sizes than in the deeper part. Furthermore, higher $\gamma(s_p)$ value near the surface might be related to the presence of roots and pebbles with various dimensions in the shallow region that could induce a strong heterogeneity in the medium.

The geostatistical fields are generated following the theoretical TPV model fitted at each depth, and each generated geostatistical field reproduces the variable $\omega$ corresponding to the normalization of $w+\epsilon$. The white noise $\epsilon$ is added to the

residual $w$ to take care of the uncertainty on $v_p$. $\epsilon$ is estimated from the 104 different velocity tomography computed with distinct inversion configurations. $\epsilon$ has a Gaussian distribution with zero mean and a variance equal to the variance of the $\log 10\left(v_p\right)$ distribution.

The random fields constitute 3D blocks of 25 m depth and are created with the Geostatistical Software Library (GSLIB; Deutsch and Journel, 1998) updated with additional libraries to compute the TPV Gaussian law (Neuman et al., 2008). GSLIB

is a collection of geostatistical programs developed to build variograms, apply kriging and generate stochastic simulations (Deutsch and Journel, 1998). The quality of the simulations was checked by looking at the distribution of the simulated residuals (Gaussian distribution with zero mean and prescribed variance) and by computing the variograms of the generated fields (see Fig. 7). The simulations were also verified by removing one by one each SRT profile to compare the distribution of the generated velocity with the removed one. Vertical cross-sections of $v_p$ parallel to profile 2 extracted from the generated

3D blocks are illustrated in Fig. S6 together with a map showing their respective locations.

**4.3 Underground structure of the whole catchment**

With $v_p$ thresholds of 700 m/s (2000 m/s), we obtain the distribution of the soil (saprolite) thicknesses on the whole catchment from the 3D $v_p$ blocks (Fig. 9). The average and standard deviation of the soil and saprolite thicknesses, computed from the 250 geostatistical models, reproduce the zonation division (Fig. 9). As expected, zones 1 and 4 share similar characteristics

with soil and saprolite thicknesses of $1.4 \pm 0.5$ m and $4 \pm 1$ m, respectively (Fig. 9). In zone 2, the soil thickness increases to $2 \pm 0.8$ m, while in zone 3, it reaches $3.4 \pm 1.1$ m. The saprolite is the thickest in zone 2, where its thickness reaches $12 \pm 1.4$ m, while it is $8.3 \pm 1.4$ m in zone 3. The deduced structure in each zone can then be used to delimit the compartment interfaces in the hydrological model NIHM. The zonation methodology we apply induces sharp contrasts of soil and saprolite thicknesses along the zones' boundaries. They reflect the strong lateral $v_p$ variations observed notably along the

SRT profiles 2 and 3.

In the following, we use the SRT data to define the thickness of the aquifer layers used in a hydrological model. We examine then how the estimated thickness uncertainty influences some of the models' outputs: piezometric and MRS data distributed over the catchment. We chose these two variables because one is representative of the saturated zone (piezometric level) while MRS also includes information of the water content in the unsaturated zone. Both simulated data types are estimated at the

same location to allow a comparison of their sensitivities. They are located at the same place where field MRS data were acquired. The location zone of those stations, their respective distance with their closest SRT profile and their Topographic Wetness Index (TWI, defined in appendix B) are summarized in Table 3.

The uncertainty on the layers' thicknesses is related to the uncertainty of the SRT data, their conversion in velocities $v_p$, the interpolation of $v_p$ over the whole catchment and to the unknown $v_p$ threshold values used to define the interfaces between

layers. Uncertainties related to the SRT data inversion and to the $v_p$ interpolation have been handled in the geostatistical framework described above. The selected $v_p$ threshold values correspond to likely values encountered in the literature

(Begonha and Bragga, 2002; Olona et al., 2010; Befus et al., 2011; Holbrook et al., 2014) and are in the value ranges estimated when comparing the SRT profiles with the field observations (Table 2). We investigate the impact of the soil bottom location by testing $v_p$ threshold values of 500, 700 and 900 m/s, keeping a fixed $v_p$ threshold at 2000 m/s to define the saprolite interface (Fig. 10a). Alternatively, we look for the influence of the saprolite bottom interface depth with $v_p$ threshold values of 1500, 2000 and 2500 m/s, the soil bottom location being defined with a 700 m/s $v_p$ threshold (Fig. 10b). The choice of those values is justified by the bibliographic analysis described in section 3.2.

From the obtained geometries, we estimate the average thickness under each MRS or piezometric stations for each applied $v_p$ threshold (Fig. 10). The generated fields correctly reproduce thicker soil under stations located in zone 3 (stations 3 and 7, Fig. 10a) and thicker saprolite in zones 2 and 3 (stations 5, 8, 22; 3 and 7, Fig. 10b) with respect to zone 1 and 4 hosting the other stations. In zones 1, 2, and 3, the soil thickness difference is higher than 1 m when comparing interfaces corresponding to distinct $v_p$ threshold values (Fig. 10a). In zone 4, that difference is less than 1 m. The soil thickness standard deviation is globally in the same order of magnitude as the average thickness difference between distinct $v_p$ thresholds. The thickness difference between $v_p$ thresholds in the saprolite is larger, with an estimated thickness difference higher than 3 m in zones 2 and 3 (stations 3, 5, 7, 8, 22; Fig. 10b). This is slightly above the standard deviation values of the thickness lower than 2 m in such zones.

## 5    Hydrological model and outputs

### 5.1 The Normally Integrated Hydrological Model – NIHM

The Normally Integrated Hydrological Model (NIHM) is a physically-based model that computes water flows by coupling processes occurring at the surface (1D stream flow and 2D surface flow) and in the subsurface compartments of a water catchment. Meteorological forcing data such as precipitations, evapotranspiration and temperatures are required NIHM inputs. We describe below the main characteristics of NIHM. A detailed description of the model and its numerical aspects are provided in Pan et al. (2015) and Jeannot et al. (2018).

The surface flow (1D and 2D) is computed through a simplified formulation of the St-Venant equations, the diffusive wave model, neglecting the inertial effects (Panday and Huyakorn, 2004). Henderson (1966) consider inertia terms to be negligible in most cases and Ahn et al. (1993) argues that such a simplification induces errors between 5% and 10% that can be treated as negligible in comparison with uncertainties on the meteorological forcing or on the hydrological data. For our application, the option that manages the diffuse 2D surface run-off and exfiltration is switched off as such processes have never been

evidenced at the Strengbach catchment. The soil covering the catchment is generally sandy, so it favors rapid infiltration even over steep slopes (Pierret et al., 2018).

The diffusive wave formulation writes:

$$\begin{cases} \dfrac{\partial A}{\partial t} + \dfrac{\partial}{\partial x}\left(-\zeta(h_r)\dfrac{\partial h_r}{\partial x}\right) = q_L - \varsigma\left(h_r - h_s\right) \\ \zeta(h_r) = \dfrac{1}{n_{GM}}\dfrac{A^{5/3}}{P^{2/3}}\left|\dfrac{\partial h_r}{\partial x}\right|^{-1/2} \end{cases}$$

(3)

The flow cross-sectional area $A$ [L2] and the wetted perimeter $P$ [L] both depend on the stream geometry. The Gauckler-Manning coefficient $n_{GM}$ [T/L1/3] is fixed at a value of $0.15\ \text{s.m}^{1/3}$. $q_L$ [L2/T] is the lateral inflow and the term $\varsigma\left(h_r - h_s\right)$ [L2/T] models the surface-subsurface coupling assuming that the exchanged water fluxes between the compartments are proportional to the head gradients between them. $h_r$ [L] is the free surface elevation and the water level $h_s$ [L] is defined by:

$$h_s\left(\mathbf{x},t\right) = \begin{cases} h\left(\mathbf{x},t\right) & \text{if } \quad h \geq z_r \\ z_r\left(\mathbf{x}\right) & \text{if } \quad h < z_r \end{cases}$$

(4)

where $h$ [L] is the groundwater head and $z_r$ [L] the riverbed elevation. Initial conditions are defined by initial values of the free surface elevation. Boundary conditions are of Dirichlet or Neuman type. At the outlet, it is assumed that the head gradient is equal to the river bed slope (flow parallel to the river bed also called zero depth gradient).

In the subsurface compartment, we assume that the water flux perpendicular to the substratum is negligible compared to the water flux parallel to the substratum. In other words, we assume that the head is constant along the perpendicular to the substratum. Following this assumption, the 3D Richards' equation is integrated (averaged) over that direction to obtain a 2D flow model. This workaround allows a significant reduction of the meshing effort, the required memory space and the computational cost while preserving the main physics of the flows (Weill et al., 2017; Jeannot et al., 2018). Comparisons with other hydrological models on benchmarks have shown that this assumption is valid (Pan et al., 2015; Jeannot et al., 2018; Weill et al., 2017).

The mathematical model of the subsurface compartment writes:

$$
\begin{cases}
\dfrac{\partial \bar{\theta}\left(h\right)}{\partial t} + \bar{S}\dfrac{\partial h(\mathbf{x},t)}{\partial t} - \nabla\cdot\bar{\mathbf{T}}\nabla h\left(\mathbf{x},t\right) = f\left(\mathbf{x},t\right) + \varsigma\left(\mathbf{x}\right)\left(h_r\left(\mathbf{x},t\right) - h_s\left(\mathbf{x},t\right)\right) & \\
h(\mathbf{x},0) = h_0\left(\mathbf{x}\right) & \mathbf{x}\in\Omega & \\
h(\mathbf{x},t) = h_D(\mathbf{x},t) & \mathbf{x}\in\partial\Omega_D & t\in\left[0,\tau_s\right] \\
\bar{\mathbf{T}}\nabla h\left(\mathbf{x},t\right)\cdot\mathbf{u} = q_N\left(\mathbf{x},t\right) & \mathbf{x}\in\partial\Omega_N & t\in\left[0,\tau_s\right]
\end{cases}
\tag{5}
$$

and

$$
\begin{cases}
\bar{\theta}\left(h\right) = \displaystyle\int_{z_w}^{z_s}\theta\left(h\right)dz & \\
\bar{S}\left(h\right) = S\left(z_w - z_b\right) & \\
\bar{\mathbf{T}}(h) = \displaystyle\int_{z_b}^{z_s}\mathbf{K}\left(h\right)dz = \int_{z_b}^{z_w}\mathbf{K}_s\,dz + \int_{z_w}^{z_s}\mathbf{K}_s k_r\left(h\right)dz
\end{cases}
\tag{6}
$$

where $\theta$ [-] is the water content, $S$ [-] the storativity and $\mathbf{T}$ the transmissivity tensor [L2T-1], the latter depending on the groundwater head. $k_r$ is the relative hydraulic conductivity, $\mathbf{K}$ [LT−1] and $\mathbf{K_s}$ [LT−1] represent the hydraulic conductivity tensor and the hydraulic conductivity tensor at saturation respectively. For our application, we consider that those tensors are isotropic, so they are reduced to the scalar values $K$ and $K_s$, respectively. $z_b$ [L] is the substratum elevation, $z_w$ [L] the groundwater free surface elevation and $z_s$ [L] the soil surface elevation. In (5), $f$ [LT−1] is the sink–source term including groundwater and the last term describes the exchange with the river. $\Omega$ is the model domain; $\partial\Omega_D$ and $\partial\Omega_N$ are partitions of the domain boundaries $\partial\Omega$ that correspond to Dirichlet and Neumann conditions, respectively. $\mathbf{u}$ is the unit vector normal to the boundary, counted positive outward. $h_D\left(\mathbf{x},t\right)$ is the prescribed head value at the Dirichlet boundaries, $q_N\left(\mathbf{x},t\right)$ is the prescribed flux at the Neumann boundaries, $h_0\left(\mathbf{x}\right)$ represents the initial conditions defined over the domain and $\tau_s$ is the simulated period.

For each element of the catchment model and at each observation time, NIHM provides the water pressure $\psi = h - z$ [L] and estimates of $\theta$ and $K$ based on the van Genuchten model for the water retention (van Genuchten, 1980):

$$
S_e(\psi) = \frac{\theta(\psi) - \theta_r}{\theta_s - \theta_r} = \begin{cases} \left(1 + \left|\alpha\psi\right|^{\eta}\right)^{-\mu} & \psi < 0 \\ 1 & \psi \geq 0 \end{cases}
\tag{7}
$$

and the Mualem model (Mualem, 1976) for the relative hydraulic conductivity $k_r$ :

$$k_r(S_e) = \frac{K}{K_s} = \begin{cases} \sqrt{S_e}\left[1-\left(1-S_e^{1/\mu}\right)^{\mu}\right]^2 & \psi < 0 \\ 1.0 & \psi \geq 0 \end{cases}$$

(8)

where $S_e$ [-] is the effective water saturation, $\theta_r$ [-] and $\theta_s$ [-] the residual and saturated volumetric water content respectively. $\alpha$ [L-1] (air entry pressure) and $\eta$ [-] are the Mualem van Genuchten shape parameters, and $\mu = 1 - 1/\eta$. In the following, we run NIHM with different values of $\theta_s$, $K_s$ and $\alpha$ but we fix the values of $\theta_r = 0.01$ and $\eta = 2$ s their influence on the stored groundwater is lesser. The three dimensional distribution of the water content can be computed by NIHM through post-processing, using the constant head assumption (since the head is assumed to be constant perpendicular

to the substratum) and (7). Water contents can then be used to estimate MRS signals at given stations as described in the next sub-section.

The equations are solved with a fully implicit non-conforming finite element method that allows a high flexibility of the discretization and ensures continuity of the normal component of the velocity from one element to the adjacent one. Although the subsurface flow model is 2D, it requires an explicit description of the parameters in three dimensions. The computation of

420 the integrals in (5) is based on the elevation and slope of the aquifer's substratum. In this paper, this geometry is estimated through seismic refraction data. The medium is divided in two vertical layers: the soil and the saprolite. The thickness of those compartments vary from a mesh to another but we consider that $\theta_s$ and $K_s$ are homogeneous in each layer.

The model has already been applied to the Strengbach catchment and showed its capacity to reproduce the behavior of the catchment flows (Pan et al., 2015). NIHM was also used to constrain the distribution of the flow lines in the Strengbach

catchment (Ackerer et al., 2020) and to explore the variability of the water transit times through the watershed (Weill et al., 2019). The comparison between observed MRS data and NIHM deduced MRS estimates was performed on the Strengbach catchment for conditioning NIHM's thickness and $\theta_s$ (Lesparre et al. 2020a).

The equations defining the groundwater flows show that key hydraulic variables such as the transmissivity $\overline{T}$ and the water content $\overline{\theta}$ correspond to the integration over the porous media thickness of the hydraulic parameters $K(h)$ and $\theta(h)$,

respectively as stated in (6). Thus, to solve the inverse problem seeking the hydrological model parameters, misestimating the the soil and saprolite thicknesses of the hydrological model underground compartments would inherently lead to a wrong assessment of the hydraulic parameters. The porous media thickness might then be considered as a sought parameter or at least as a prior information associated with an uncertainty. All the more, measurable data sensitive to $\overline{T}$ and $\overline{\theta}$ should be completed

with data directly related to the porous media thickness to tackle the porous media thickness correlation with $K(h)$ and $\theta(h)$ in (6).

## 5.2 MRS data estimate

MRS is a non-invasive geophysical method that is classically used to estimate the underground water content in the saturated and unsaturated zones of the subsurface (Legchenko et al., 2004; Costabel and Günther, 2014; Mazzilli et al., 2016). Thirty-two MRS measurements were performed on 23 different stations covering the Strengbach catchment during two campaigns in April and May 2013. Data were acquired with a Numis plus device system from IRIS instruments using eight-shaped square loops. This data set was fully described in Lesparre et al. (2020b). A first analysis of the MRS measurements described the subsurface water content distribution over the catchment (Boucher et al., 2015; Pierret et al., 2018). A subset of the data acquired at 16 stations was then used as a posterior information to select subsurface parameters of NIHM applied on the Strengbach catchment (Lesparre et al., 2020a). Here, we estimate MRS synthetic data from NIHM simulations. The MRS signal envelope $V(q,t)$ decays with time $t$ during the sounding for a pulse moment $q$. It can be written as follows (Legchenko and Valla, 2002):

$$V(q,t) = \int_z \kappa(q,z) \cdot \theta(z) \cdot \exp\left(-t / T_2^*(z)\right) dz$$

(9)

where $\kappa(q,z)$ represents the kernel function of the MRS vertical sensitivity and depends on the geometry of the acquisition system and the amplitude of the injected pulse $q$. $\kappa(q,z)$ is influenced by environmental conditions such as the geomagnetic field amplitude, the Larmor frequency and the electrical resistivity of the subsurface (Legchenko and Valla, 2002). The values of the parameters used for the computation of $\kappa(q,z)$ are given in Lesparre et al. (2020b). The shape of $\kappa(q,z)$ is defined by the geometry of the vertical layers whereby the water content $\theta(z)$ and the relaxation time $T_2^*(z)$ are provided by NIHM. Here, as we work with synthetic MRS signals, we assume that $\kappa(q,z)$ and $T_2^*(z)$ do not vary with time. We consider $T_2^* = \mathrm{median}(T_{2\mathrm{app}}^*)$, with $T_{2app}^*$ the apparent value of the relaxation time estimated for each pulse (see Lesparre et al., 2020a). Then, we use the $\theta(z)$ values provided by NIHM to compute values of $V(q,t)$ with (7) and investigate how they evolve with the tested geometries and parameters' sets.

# 6 Impacts of layer thicknesses on hydrology variables

## 6.1 Test case setup

The influence of the soil and saprolite thicknesses on hydrological variables is analysed using two outputs: piezometric heads linked to the saturated thickness and water content (through MRS) related to the water stored in the saturated and unsaturated media. This influence is quantified by a simplified sensitivity analysis that consists in running the hydrological model NIHM for each 250 simulated velocity fields with the following input parameters: distinct velocity thresholds to define the layers' thicknesses and different sets of hydraulic parameters. We focus our investigations on testing the impact of the hydraulic conductivity $K_s$, the saturated water content $\theta_s$ and air pressure entry $\alpha$. Preliminary tests showed that the considered outputs (MRS data and piezometric heads) are mainly sensitive to those hydraulic parameters together with the thickness of the underground layers.

In a first step, we fix the set of hydraulic parameters and investigate combinations of soil and saprolite $v_p$ thresholds. We assess soil $v_p$ threshold values of 500, 700 and 900 m/s for a fixed $v_p$ threshold at 2000 m/s in the saprolite. We also examine saprolite $v_p$ threshold values of 1500, 2000 and 2500 m/s with a soil $v_p$ threshold fixed at 700 m/s. Thus, we test five combinations of $v_p$ thresholds, each shifting the soil and saprolite thickness patterns and influencing the global porous volume of the CZ underground compartments as well as their transmissivity. In a second step, we prescribe the $v_p$ thresholds to 700 m/s for the soil and 2000 m/s for the saprolite and apply three different sets of hydraulic parameters detailed in Table 4. The values given to each parameter are defined considering a previous study of the Strengbach vadose zone (Belfort et al., 2018). We note that in similar granitic catchment contexts, porosity values (that we relate to $\theta_s$) as high as 50% and 60% have been estimated in the shallow region (Holbrook et al., 2014, 2019).

Simulations are run with the meteorological forcing measured on the Strengbach catchment from June 1, 2012, to May 31, 2013, as this period covers the MRS measurement campaign. We analyze data estimated at a same date, 19th of April 2013, so we can compare data related to a same meteorological forcing history. This date corresponds to a relatively low water level and only a few artesian locations might be observed. Artesian events might indeed happen depending on the applied parameters, the $v_p$ thresholds and the station location. Because NIHM is not designed to simulate these situations properly, we prefer to focus the data sensitivity analysis to an average flow period to limit the occurence of such events and so variations of the water table can still occur.

The head levels are converted to water table depths (WTD). For MRS data, we focus the analysis on the signal simulated for the pulse that shows the largest variability when compared to the other pulses applied on the field. A high water content in the

underground induces a high MRS signal amplitude and a water level close to the surface corresponds to a low WTD value. Results are first presented on 3D plots that represent the projection of the simulations on three planes: thicknesses of both layers (horizontal plane) and MRS signal or WTD values in function of the 2 layers' thickness (vertical planes; Fig. 11 and 12). When exploring the influence of the parameters' set, data on the horizontal plane are in grey since the soil and saprolite thicknesses vary with the same distribution for the three studied sets (Fig. 12). We discuss data estimated at stations 5 and 6 which are representative of the main results. Results of all stations are given in supplementary materials (Fig. S7 to S10). Stations 5 and 6 differ in soil thickness (less than 3 m for station 6, less than 5 m for station 5) and in saprolite thickness (between 1 m and 12 m for station 6, and 5.5 m to 16 m for station 5). Therefore, the total thickness of the aquifer below those stations is not similar (Fig. 10). Station 5 represents zones where the topography favors water storage (high TWI) whereas the topography is propitious to water drainage around station 6 (low TWI).

We then estimate the coefficient of determination of a linear regression $R^2$ between the estimated data and the soil or saprolite thicknesses for all stations to describe how their specific location influences the data sensitivity to the thicknesses variations (Fig. 13 and 14). $R^2$ highlights a linear relationship between the estimated data and the layer thickness when it is close to 1. However, a coefficient significantly different from 1 does not mean that the data are not dependent on the layer thickness.

Stations 9, 13, and 14, are located less than 10 m from a seismic profile (Table 3), therefore measured $v_p$ values strongly constrain the soil and saprolite thicknesses that are accurately estimated for given velocity thresholds (Fig. 10, S7 to S10). Those narrow variation ranges hinders analyzing the correlation between the soil and saprolite thicknesses and the estimated data so we do not include such stations in the $R^2$ analysis.

Contrarily to other geophysical methods, MRS is directly sensitive to the underground water content as no petrophysical relationship is required to estimate the MRS signal from water contents estimated by a hydrological model. However, the signal measured on the field is impacted by the instrument dead time, the pulse length and the presence of bounded water cannot be detected. In the analysis applied to synthetic estimates, we did not consider such aspects that influence MRS measurements in addition to the hydraulic parameters' values. They should be taken under consideration in the analysis of real MRS data.

**6.2 Groundwater variations with respect to the porous medium thickness**

For a given set of parameters (e.g., set B in Table 4), we investigate the influence of the $v_p$ thresholds on the MRS and WTD values. Note that the $v_p$ threshold of the soil layer influences the saprolite thickness: the lower the soil threshold, the thicker the saprolite layer for a same $v_p$ threshold of the saprolite layer. Results clearly show the important effect of the station location on the MRS amplitude which varies in [10-100] nV at station 6 and [100-300] nV at station 5 (Fig. 11a and b). WTD

values are also strongly impacted as they vary between 1 and 15 m at station 6 and between 1 and 3 m at station 5 (Fig. 11c

and d). The thicker underground medium under station 5 and its position on a region favoring storage (high TWI) explain its

higher MRS and lower WTD values. The sensitivity of the data to $v_p$ is clearly different for these two stations. At station 6, the MRS signal is proportional to the soil thickness for small saprolite thickness (less than 2 m, black dots Fig. 11a). For higher saprolite thicknesses, the MRS signal is lower and linearly dependent on the saprolite thickness (yellow dots Fig. 11a). The WTD is linearly dependent on the thickness of the saprolite layer, since the WTD is mostly below the soil layer (Fig. 11c). At

520 station 5, MRS estimates obtained with all $v_p$ thresholds show a linear trend with the soil thickness, but none of them show such a trend with the saprolite thickness (Fig. 11b). WTD values do not show any linear dependence on the soil or saprolite thicknesses (Fig. 11d). However, a thicker soil layer is related to a deeper WTD and high MRS values (green dots, Fig. 11d). A thicker soil provides more space to store water leading to a stronger MRS amplitude, but it also increases the transmissivity that might favor drainage and thus reduce the water level. Fig. 11 also highlights non-uniqueness of MRS and WTD with

525 respect to the geometry for a given hydrological parameter set. In particular, at station 5, a given value of WTD can be obtained by different combinations of the layers' thicknesses. It is less true for MRS at the same station where the number of possible combinations is lower due to the correlation with the soil thickness. This clearly shows the interest of using different kinds of measured variables to better constrain the model.

A global overview of the correlations that may exist between layers' thicknesses and MRS or WTD is provided in Fig. 13. For

almost all stations, when the correlation with the soil (resp. saprolite) thickness for MSR or WTD is significant, the estimated data are not linearly correlated for saprolite (resp. soil). On average, MRS with $R^2$ values above 0.5 (Fig. 13a) are more linearly dependent on the soil thickness than WTD for which $R^2$ values remain mostly below 0.5 (Fig. 13b). WTD is more controlled by the saprolite thickness as $R^2$ values above 0.5 are observed (Fig. 13d). This can be explained by the fact that WTD depicts the water level of the saturated medium that might remain in the saprolite under dry conditions, while MRS

depends on the water content variations in both the saturated and unsaturated media. In general, MRS and WTD better correlate with the soil thickness when the saprolite is thinner (black plus lines Fig. 13). On the contrary, both data types better correlate with the saprolite thickness when it is thicker (yellow star lines Fig. 13). A thicker saprolite hinders the presence of water in the soil as the water level might be lower, and also since it increases the transmissivity and thus favors drainage. Therefore, the influence of the soil thickness on the estimates is annihilated. In contrast, a thin saprolite is more likely saturated by a

higher water level and a reduced transmissivity so its thickness influence on the estimates diminishes. For both data types, stations with a low TWI are generally better correlated with the saprolite thickness than stations with a high TWI. A low TWI indicates a region favorable to drainage and thus to a low water level for a given aquifer bottom, therefore the groundwater level is more likely present in the saprolite.

Stations 3 and 7 show lower $R^2$ values compared with stations characterized with a similar TWI, in particular for MRS (WTD) values compared with the saprolite (soil) thickness (Fig. 13b and c). Those stations located in zones 3 present thicker soil and saprolite (Table 3 and Fig. 10). The high WTD under those stations does not help exploring the influence of the soil thickness (Fig. S8). WTD and MRS at stations 5, 8, and 22 seem to be independent from the saprolite thickness. Those stations associated to a relatively high TWI are located in zone 2 (Table 3) that is relatively flat and thus propitious for water storage. The WTD underneath those stations is close to the surface and varies in a range of 1 m or less indicating that the WTD is not strongly influenced by the variability of the layers' thickness (Fig. S8). MRS is still strongly correlated to the soil thickness under stations 5 and 8 as MRS depends on the water content in the unsaturated medium and the water table is between 1 and 2 m below the surface at those stations (Fig. S7). However, at station 22 with the highest TWI value, the water table is very close to the surface, when not in artesian conditions (Fig. S8), MRS or WTD cannot be influenced by the underground medium thickness for our parameter sets.

## 6.3 Groundwater variations with respect to the hydraulic parameters

We investigate now the effects of the hydraulic parameters for a given set of $v_p$ thresholds of 700 m/s for the soil and 2000 m/s for the saprolite (Fig. 12). Thicknesses variations of the soil and saprolite are thus reduced since they are only related to the generation of the 250 geostatistical models. Despite that, we note that the range of variations of both signals are similar as in the previous test. Again, non-uniqueness occurs as different parameter sets give the same MRS or WTD values for given $v_p$ thresholds. However, the relationship between both data types and layers' thicknesses is parameter set dependent. This is clearly shown for MRS and soil thickness at station 5 (Fig. 12b) and for WTD and saprolite thickness at station 6 (Fig. 12c). At station 6, we observe lowest WTD values for the parameter set C that corresponds also to the highest MRS signal reflecting high saturated conditions (yellow dots Fig. 12 a and c). The parameter set C has the lowest saprolite $\theta_s$ and so a smaller storage capacity compared to the two other parameters' sets. This small storage capacity leads to a higher water level in the medium below the station. Set C corresponds also to saprolite layers with the lowest $K_s$ that further induces a slower drainage and thus might better maintain the groundwater under the station. All the more, the set C shows the highest $\theta_s$ of the soil which provides a larger space to store water in the unsaturated zone and induce a higher MRS signal.

At station 5, WTD values corresponding to parameter A (blue dots) are slightly higher than values estimated with the other parameters (Fig. 12d). However, if the parameter set influences the trend between MRS estimates versus the soil thickness, we do not distinguish a clear impact of the parameter set on the MRS signal amplitude (Fig. 12b) as observed at station 6. This means that above a given aquifer thickness, variations of the WTD due to distinct sets of parameters influence less the MRS signal than variations of the soil thickness.

Influence of the parameter sets on the correlations that may exist between layers' thicknesses and MRS or WTD for all stations is illustrated in Fig. 14. In general, MRS and WTD better correlate with the soil (resp. saprolite) thickness for parameter set C

(resp. A). Thus, low values of $\theta_s$ and $K_s$ in the saprolite associated with a high $\theta_s$ in the soil that characterizes parameter set C lead to a better correlation of the estimated data to the soil. Such parameters favor a water table closer to the surface and a higher water storage, both allowing a stronger influence of the soil thickness on the WTD and MRS. In contrast, high values of $\theta_s$ and $K_s$ in the saprolite of the parameter set A induce a better drainage and thus lead to a lower water level. In that case, WTD and MRS are more sensitive to the saprolite thickness. Stations 3, 7, 8, 5 and 22 show a general lower sensitivity to the

medium thickness as mentioned in section 6.2. Here again, this peculiar behavior can be explained by a thicker medium, with higher TWI values for stations 8, 5 and 22.

The complementarity between the piezometric heads and the MRS is again emphasized as both signals are differently influenced by the parameters tested. It is yet difficult from the results we obtain to distinguish what is the respective influence of those parameters on the synthetic data.

**6.4 Insights from the test case**

The structure of the CZ underground compartments can be deduced from geophysical images of the subsurface obtained classically along profiles. By analyzing the geostatistical variations of such images, it is then possible to provide an insight of the subsurface geometry beyond the geophysical profiles. Assuming that patterns of hydrological facies can be deduced from variations of the geophysical properties, it is then possible to build the geometry of distributed hydrological models at the

catchment scale from a few geophysical profiles. Here we use SRT data to define the interface geometry between layers with distinct hydrological properties (Befus et al., 2011; Flinchum et al., 2018; Pasquet et al., 2022). The interpolation methodology used to design the geometry of a hydrological model from such data could be applied to other catchments displaying a heterogeneous thickness distribution of its subsurface compartments. Similarly, Electrical Resistivity Tomography (ERT) could be used in catchments characterized by relatively homogeneous and/or layered compartments. Yet the occurrence of a

variable clay component at the catchment scale, as observed on the Strengbach, further complicates the interpolation of ERT data. Furthermore, the relation between the geophysical properties and the subsurface structure requires field measurements to sustain the generalization of boundary delineation between the distinct hydrofacies. Our basic sensitivity analysis explores how measurable data as WTD and MRS are impacted by the variability of the surface compartments' thicknesses related to the geostatistical uncertainty and to the choice of the $v_p$ thresholds. We show that this sensitivity is influenced by the values

of some key hydraulic parameters (namely the water content at saturation and the hydraulic conductivity at saturation).

It appears from this analysis that MRS is differently sensitive to the subsurface properties than WTD. In particular, MRS shows a better sensitivity to the shallowest subsurface compartment under higher drainage conditions. The drainage capacity of the subsurface can be related to 1) the topography as slopes favor drainage; 2) the aquifer thickness since a thicker aquifer leads

to a higher transmissivity for a same hydraulic conductivity and 3) the hydraulic parameters as obviously a higher hydraulic
conductivity induces a better drainage. Conspicuously, in well drained areas where the WTD does not reach the soil, its measurement does not supply information on the properties of that superficial medium. In contrast, MRS detects the quantity of unbounded water and we show here that MRS presents globally a higher sensitivity to the shallowest compartment thickness, the soil, in comparison with WTD. MRS acquisition protocols adapted to sound the superficial media should then supply complementary information to WTD with the aim of constraining the hydraulic parameter values (Costabel and Günther,
2014). MRS could then be pertinent to calibrate hydrologic models at the catchment scale notably when acquired along draining slopes and in places characterized with a deeper porous medium. In areas where it is crucial to understand the conditions of recharge, MRS data would then be of great interest.

The test case shows the influence of the hydraulic parameters on the MRS and WTD sensitivity to the subsurface compartments' thicknesses. A global sensitivity analysis should help discriminate the impact of the compartment thicknesses
and different hydraulic parameter values on those data. We focused our analysis at a given time, while the variations of the water content in the underground evolve depending on the forcing conditions. It would therefore be interesting to extend the sensitivity analysis to a longer time range in order to assess the impact of the hydrological regime on the data sensitivity.

**7 Conclusions**

In this paper we propose a methodology to build the pattern of the three dimensional underground heterogeneity from
geostatistical analysis of seismic profiles acquired on the Strengbach catchment. The computation of preliminary variograms shows no vertical coherency of the seismic data, allowing a depth-by-depth analysis. The properties of the experimental variograms reflect the data uncertainty variations with depth, the spatial resolution of the SRT, and the dimension of the underground structures. The porous soil and saprolite compartments are assumed to drive most groundwater flow supplying

the catchment outlet studied here. The thicknesses of those layers are deduced by defining $v_p$ thresholds from field
observations and considering previous studies in similar contexts. The study shows that the average soil and saprolite thicknesses are thinner on the catchment crests, upper slopes, and the valley bottom close to the outlet. At the bottom of steep slopes, the largest soil thicknesses occurred together with high saprolite thickness. In a flat area upstream the creek's main spring, the soil is also relatively thick, and the saprolite appears to be the thickest.

Increasing the $v_p$ threshold globally shifts the soil and saprolite compartments' bottom limits downward. Thus, an increase in
the $v_p$ threshold is equivalent to a transmissivity rise in the different layers (Eq. 5) and an increase of the storage capacity. This tends to lower the groundwater level and induces higher WTD and lower MRS values for a given set of hydraulic parameters. The sensitivity of the WTD and MRS signal to the porous medium thickness is also depending on the set of hydraulic parameters. For instance, low hydraulic conductivity and porosity of the saprolite favor shallower groundwater levels

and higher signal sensitivity to the soil thickness. Beyond the valuable information supplied by SRT on the Strengbach catchment underground structure, this paper also shows the double dependence of data influenced by the water quantity (i.e. WTD and MRS) to both the hydraulic parameters and the thickness of the porous media. Thus, the model geometry knowledge is crucial to reduce the non-uniqueness of the hydrological inverse problem that would fit such data. SRT measurements should be completed with field observations in pits or on outcrops so they could constrain efficiently the hydrological inverse problem. The tests applied here demonstrate that piezometric heads and MRS signals display different underground structure sensitivity even when collocated. Such a complementarity is very encouraging for setting up future experiments. Data presently recorded with piezometers could be constructively completed with repeated MRS acquisitions sensitive to the medium porosity. The proposed methodology opens the way for applying hydro-geophysical measurements to constrain underground CZ structures (using SRT) and their hydraulic properties (with piezometers and MRS). This demonstrative application could be easily translated into other watersheds where MRS measurements have been or could be acquired for constraining their hydraulic parameters. The design of the SRT profiles distribution should investigate the different underground morphology susceptible to occur on the catchment. This study's field-based synthetic exploration invites a quantitative global sensitivity analysis to deepen the understanding of the respective impact on the different data types of the hydraulic parameters and their eventual combined effects.

**Appendix A**

The whole set of inverted SRT profiles is represented Fig. A1.

**Appendix B**

The topographic wetness index (TWI) helps distinguishing the capacity of a station to store or drain the groundwater depending on the geometry of the topography. The TWI depends on the upstream contributing area per unit width orthogonal to the flow direction ($a$) and on the local slope ($b$), and is defined as (Beven and Kirkby, 1979):

$$\text{TWI} = \ln\left(\frac{a}{\tan(b)}\right).$$

(B1)

A low TWI value indicates a region suitable to drainage while higher TWI values correspond to areas favoring water storage. We compute TWI values at each MRS station (Fig. 1, Table 3) to classify the obtained results and sustain the data sensitivity interpretation. The sensitivity might indeed be influenced by the spatial configuration of the measurement stations that strengthens a groundwater drainage or storage behavior.

## Data availability

The whole seismic data set is available on the H+ database (http://hplus.ore.fr/en/; Pasquet et al., 2019), which stores the geophysical data collected on the CZ observatories of the OZCAR network. The fortran libraries developed to run the NIHM code are available upon request to the authors. The python library used to analyze the SRT data can be downloaded at www.pygimli.org (Rücker et al., 2017). The geostatistical software library, GSLIB, is distributed at www.gslib.com (Deutsch and Journel, 1992).

**Competing interests.** The authors declare that they have no conflict of interest.

## Acknowledgements.

The authors warmly thank Marie-Claire Pierret, Hélène Jund, Marie-Anne Churka, Quentin Chaffaut, Flore Rembert, Sylvain Weill, Benjamin Belfort, Matthias Oursin, Jérôme Vergne, and Sylvain Benarioumlil for their help in the acquisition of the seismic data presented hereThe digital elevation model was obtained from aerial LiDAR images recorded in 2018 by the Helimap system. Computing time was provided by the HPC-UdS. We thank four anonymous reviewers and Jacopo Boaga for their constructive remarks that help us improve the quality of the manuscript.

## Financial support.

Meteorological and flow rate data collection was funded by the Observatoire Hydro-Géochimique de l'Environnement (OHGE), which is financially supported by CNRS/INSU France and the University of Strasbourg. OHGE is part of the OZCAR research infrastructure (http://www.ozcar-ri.org). . The CRITEX ANR-11-EQPX-0011 project provided the instruments used for the seismic data acquisition, and the ANR HYDROCRIZSTO-15-CE01-0010-02 project funded the field campaigns.

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

**Table 1: Acquisition parameters of the seismic lines.**

| Line number | 1 | 2 | 3 | 9 | 10 | 11 | 12 | 13 | 14 | 15 |
|---|---|---|---|---|---|---|---|---|---|---|
| Number of traces | 144 | 144 | 144 | 96 | 96 | 72 | 72 | 96 | 96 | 96 |
| Trace spacing (m) | 2 | 2 | 2 | 2 | 2 | 2 | 2 | 2 | 2 | 2 |
| Line length (m) | 286 | 286 | 286 | 190 | 190 | 142 | 142 | 190 | 190 | 190 |
| Number of shots | 30 | 30 | 30 | 25 | 25 | 19 | 19 | 25 | 25 | 25 |
| Shot spacing (m) | 10 | 10 | 10 | 8 | 8 | 8 | 8 | 8 | 8 | 8 |
| Recording time (s) | 0.75 | 0.75 | 0.75 | 0.8 | 0.8 | 0.8 | 0.8 | 0.8 | 0.8 | 0.8 |
| Sampling time (ms) | 0.125 | 0.125 | 0.125 | 0.125 | 0.125 | 0.125 | 0.125 | 0.125 | 0.125 | 0.125 |
| Time delay (s) | -0.1 | -0.1 | -0.1 | -0.05 | -0.05 | -0.05 | -0.05 | -0.05 | -0.05 | -0.05 |

**Table 2: Depth of the Bottom Interfaces Estimated at the Boreholes and the Corresponding $v_p$ Ranges of the Closest Part of the Seismic Profiles at such Depths. The Saprolite Bottom Interface Is Not Intercepted by Pz10b.**

| Borehole name | Closest profile number | Minimum distance to the closest profile (m) | Bottom interface depth (m) | | Corresponding $v_p$ range (m/s) | |
|---|---|---|---|---|---|---|
| | | | Soil | Saprolite | Soil | Saprolite |
| F1 | 9 | 35 | 0.5 | 1.5 | 480; 720 | 900; 1030 |
| Pz3 | 13 | 13 | 1 | 4.5 | 410; 630 | 1480; 2245 |
| Pz10b | 3 | 9 | 0.5 | - | 560; 650 | - |


**Table 3: MRS Station Zone Locations and Distance to Their Closest SRT Profile.**

| MRS station number | Zone number | Closest SRT profile number | Distance to the closest profile (m) | Topographic wetness index |
|---|---|---|---|---|
| 1 | 4 | 11 | 163 | 7.6 |
| 3 | 3 | 2 | 31 | 6.6 |
| 4 | 4 | 15 | 30 | 6.3 |
| 5 | 2 | 1 | 60 | 8.0 |
| 6 | 4 | 3 | 44 | 6.2 |
| 7 | 3 | 3 | 69 | 5.8 |
| 8 | 2 | 1 | 93 | 7.5 |
| 9 | 1 | 12 | 6 | 7.6 |
| 12 | 4 | 13 | 48 | 7.2 |
| 13 | 4 | 9 | 2 | 6.3 |
| 14 | 4 | 15 | 5 | 5.8 |
| 15 | 4 | 1 | 130 | 7.7 |
| 16 | 4 | 1 | 39 | 6.7 |
| 19 | 4 | 3 | 116 | 6.5 |
| 22 | 2 | 1 | 29 | 8.5 |
| 23 | 4 | 9 | 231 | 7.4 |

**Table 4: Parameters Applied in Each of the Subsurface Compartments for the Different Sets of Simulation Runs**

| Parameter | $\theta_s$ (%) | | $K_s$ (m/s) | | $\alpha$ (m−1) | |
|-----------|------|-----------|------|-----------|------|-----------|
| Medium | Soil | Saprolite | Soil | Saprolite | Soil | Saprolite |
| Set A | 0.1875 | 0.08 | $10^{-4.5}$ | $10^{-4.5}$ | 0.575 | 1.525 |
| Set B | 0.325 | 0.06 | $10^{-4}$ | $10^{-5}$ | 1.05 | 1.05 |
| Set C | 0.4625 | 0.04 | $10^{-3.5}$ | $10^{-5.5}$ | 1.525 | 0.575 |

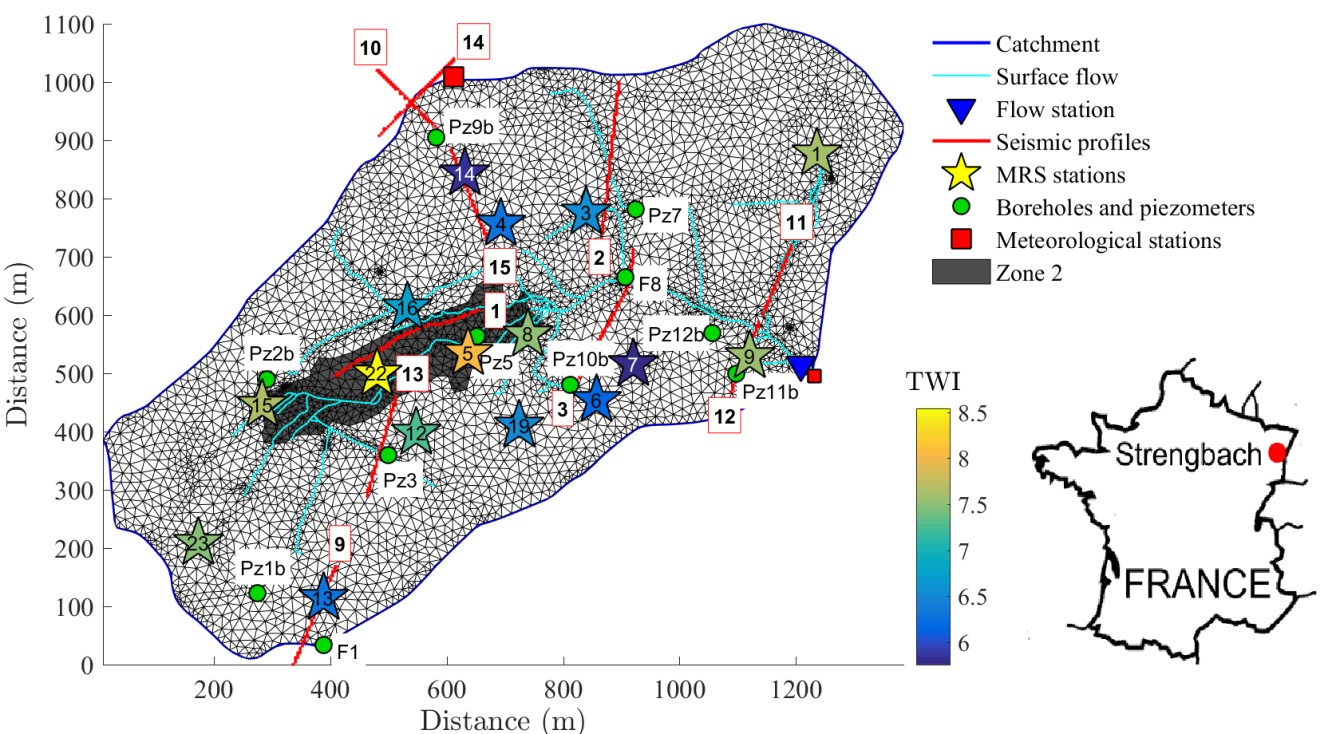

Figure 1: Map of the Strengbach catchment. The seismic profiles are indicated by red lines, and the flow measurement station at the outlet is represented by the blue triangle. The colored stars show the TWI, defined in Eq. (B1), computed at each MRS station. The white triangles represent the 2D mesh of the hydrological model underground compartment, with the grey zone 2 that was identified as a storage area in Lesparre et al. (2020a). The cyan lines represent the 1D mesh of the hydrological model surface compartment that includes flows in the creek but also on the forestry roads. An inset shows the location of the Strengbach catchment in the Northeast of France.

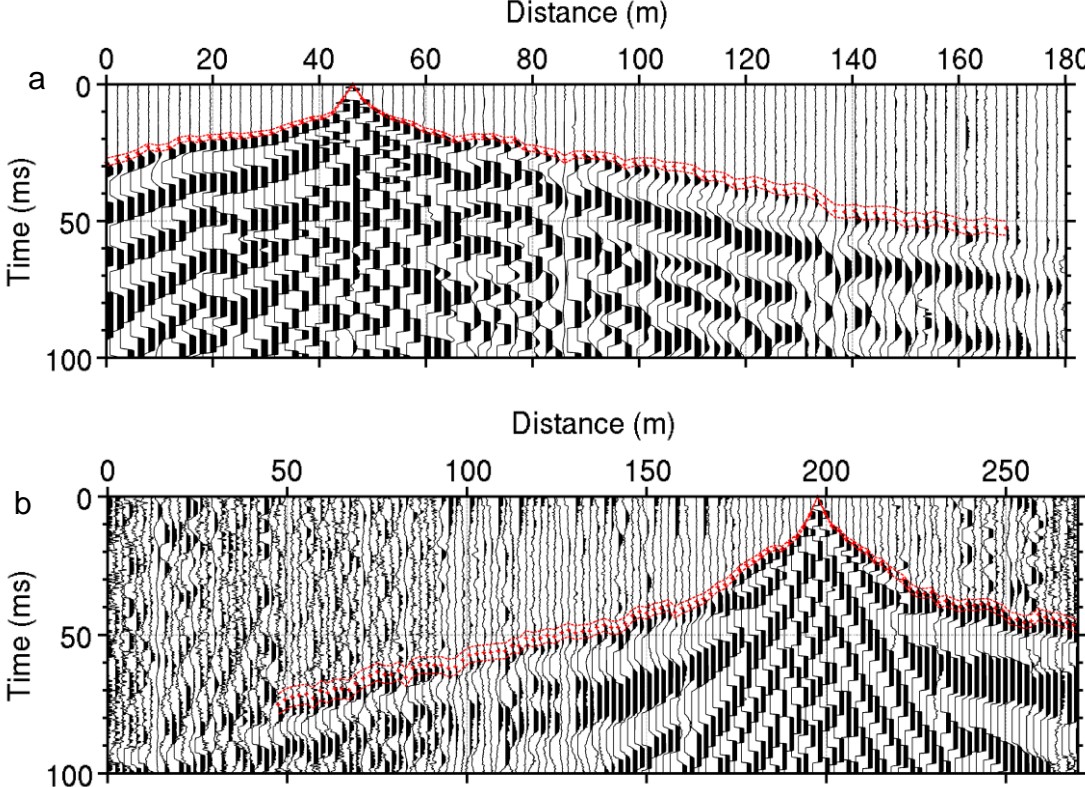

**Figure 2 Examples of shots along (a) line 15 (96 geophones) and (b) line 1 (144 geophones). The red error bars indicate the manually picked first arrival times.**

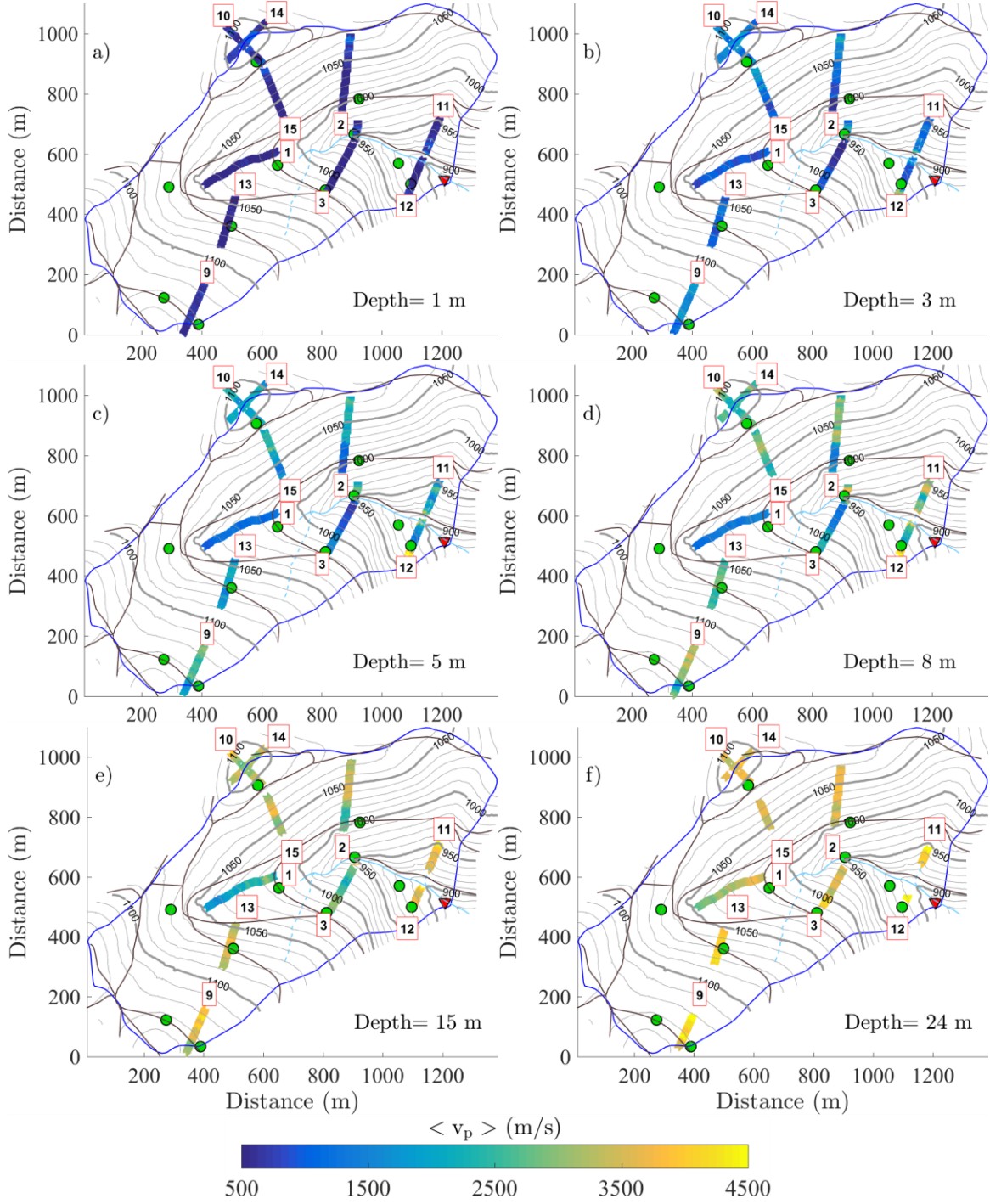

**Figure 3: Spatial distribution on the Strengbach catchment of the $v_p$ extracted at different depths from the SRT inversion profiles (Fig. A1).**

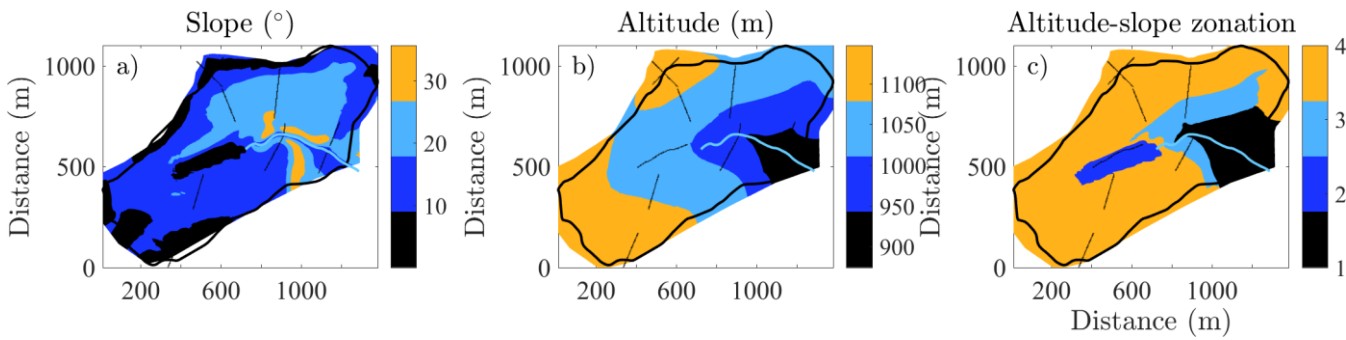


**Figure 4: Analysis of the Strengbach catchment topography to delimit zonation in which the $v_p$ presents geostationary characteristics.**

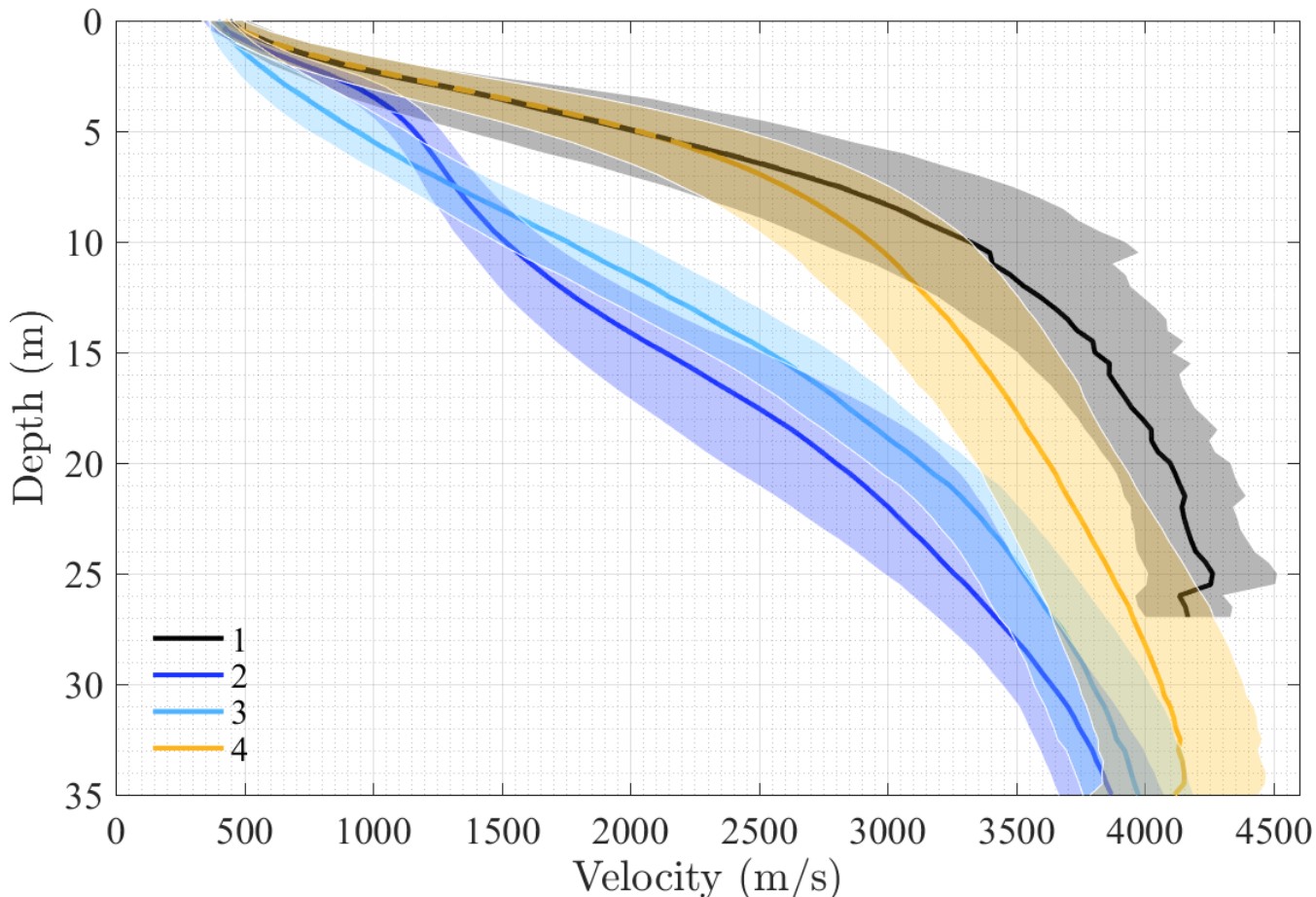

Figure 5: Average $v_p$ as a function of the depth in each zone. The shaded areas represent the average $v_p$ more or less

1 standard deviation of $v_p$ .

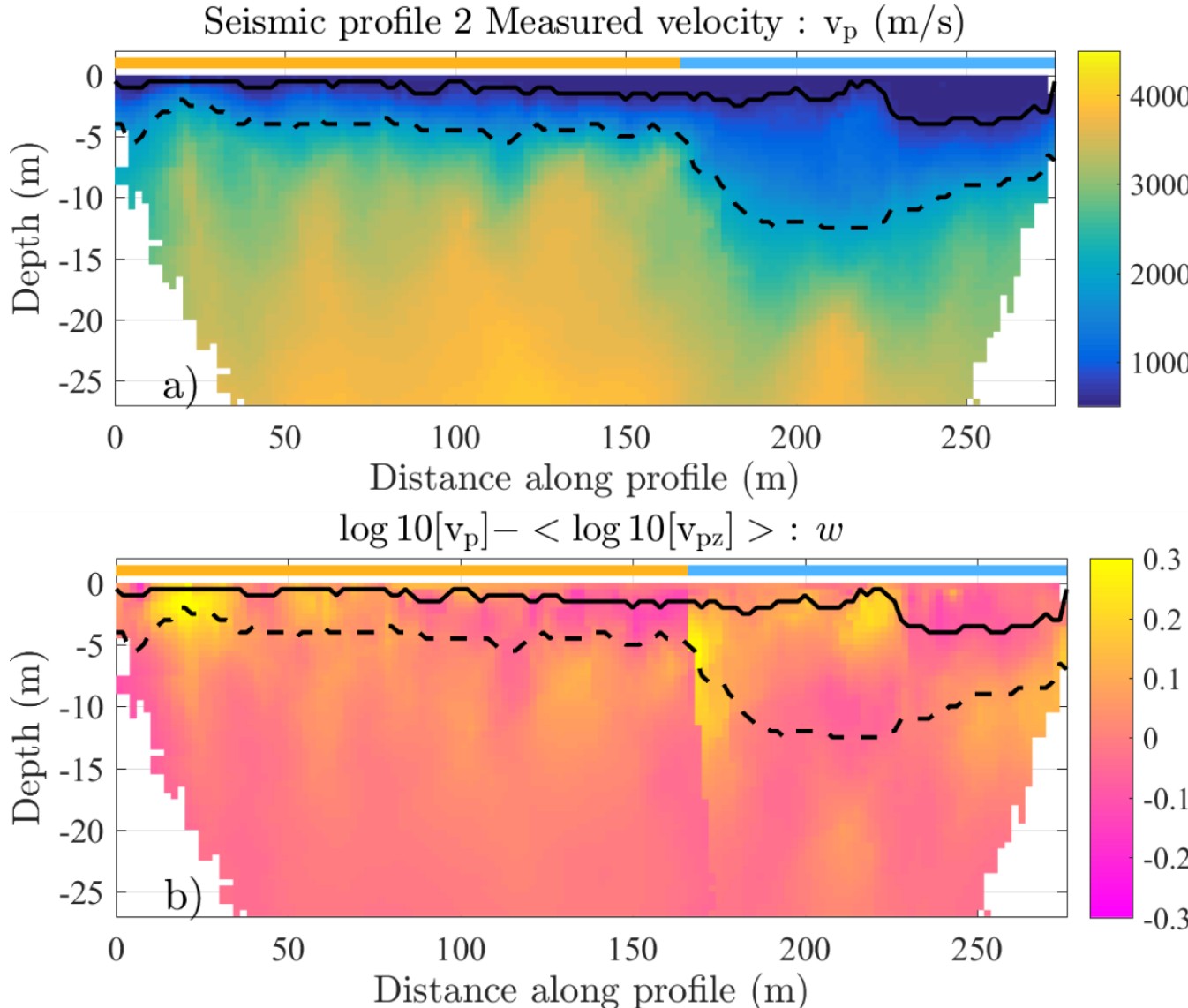

**Figure 6: Measured $v_p$ as estimated after inversion along profile 2 (a). $w$ variations after the trend removal (b). The yellow and blue clear lines above the profiles represent the profile parts that are in zones 4 and 3, respectively. The solid (dashed) black line represents the soil (saprolite) bottom interface for a $v_p$ threshold of 700 m/s (2000 m/s).**

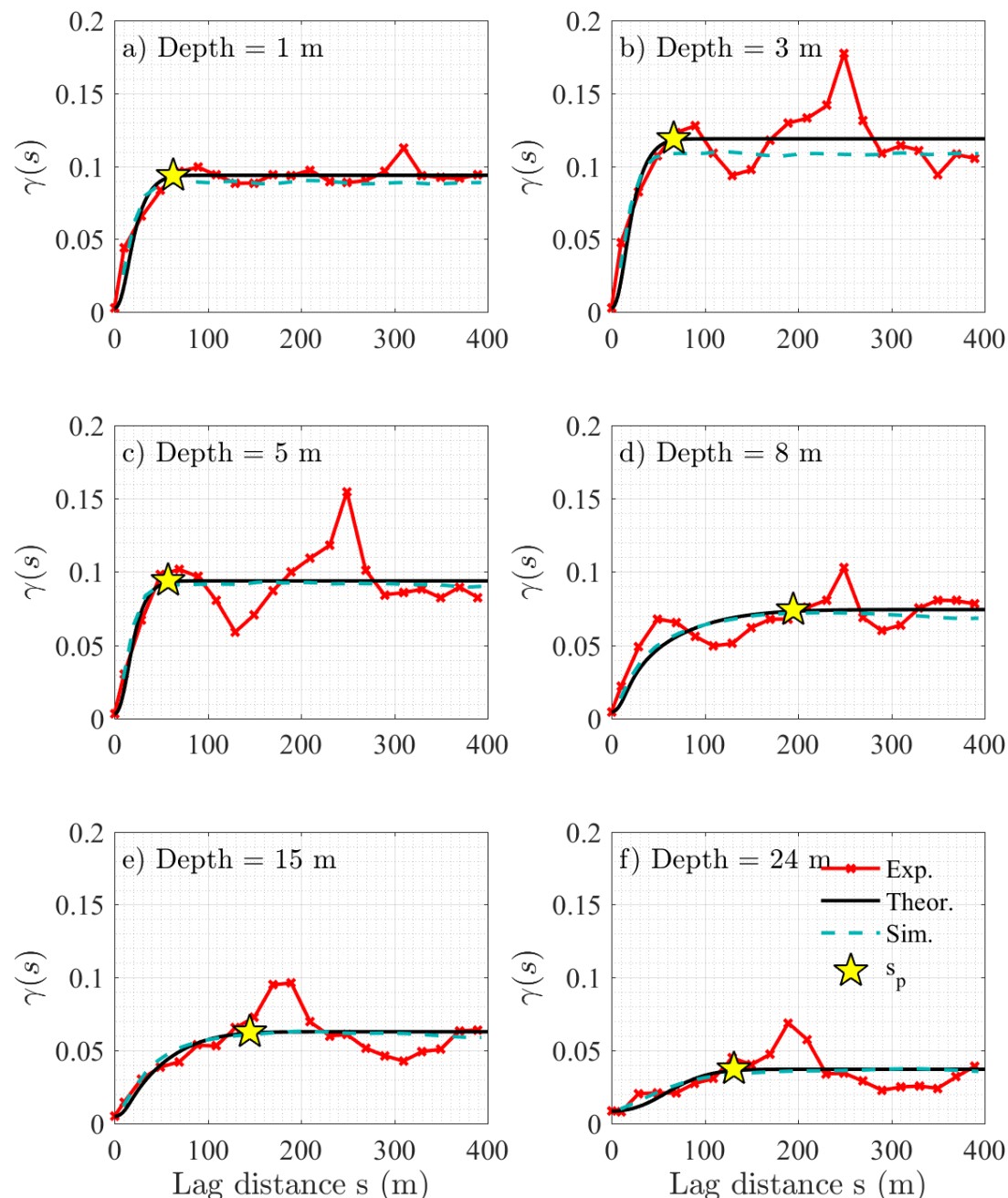


**Figure 7: Experimental variograms (red lines and crosses) estimated from the detrended variable $w$. The theoretical variograms (black lines) follow a Gaussian truncated power value law. $s_p$ (yellow star) represents the lag distance where the variograms reach a plateau. The variogram of the generated field is represented by the blue dashed line.**

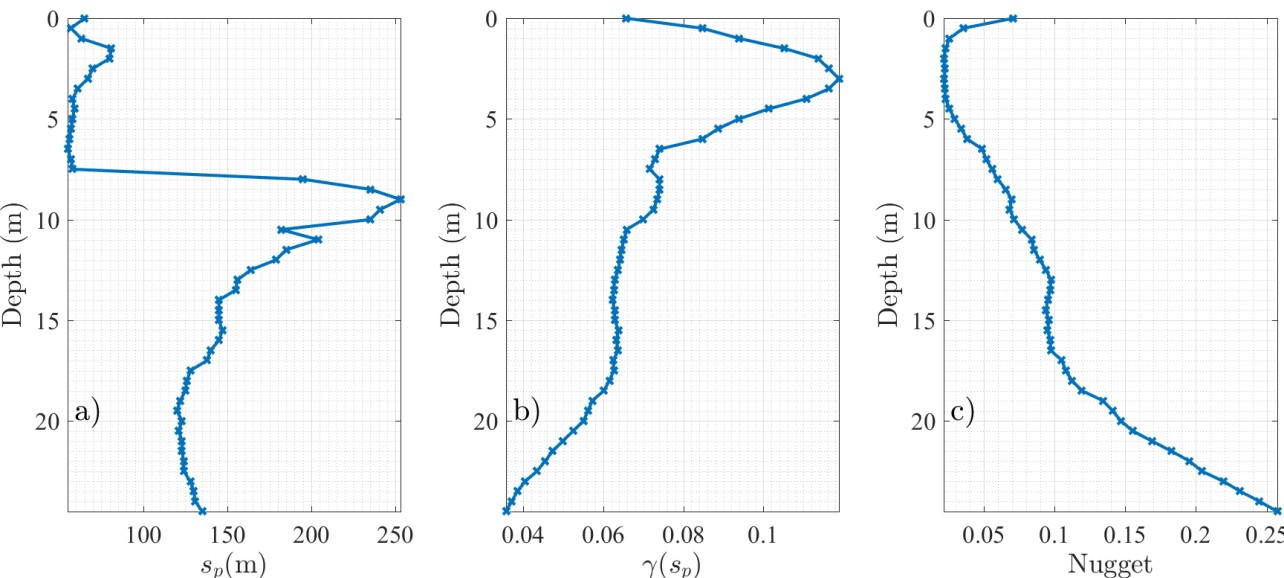

Figure 8: Characteristics of the theoretical variograms as a function of depth: $s_p$ (a), $\gamma(s_p)$ (b) (see Fig. 6). The nugget is directly fixed from the standard deviation of the SRT profiles (c).

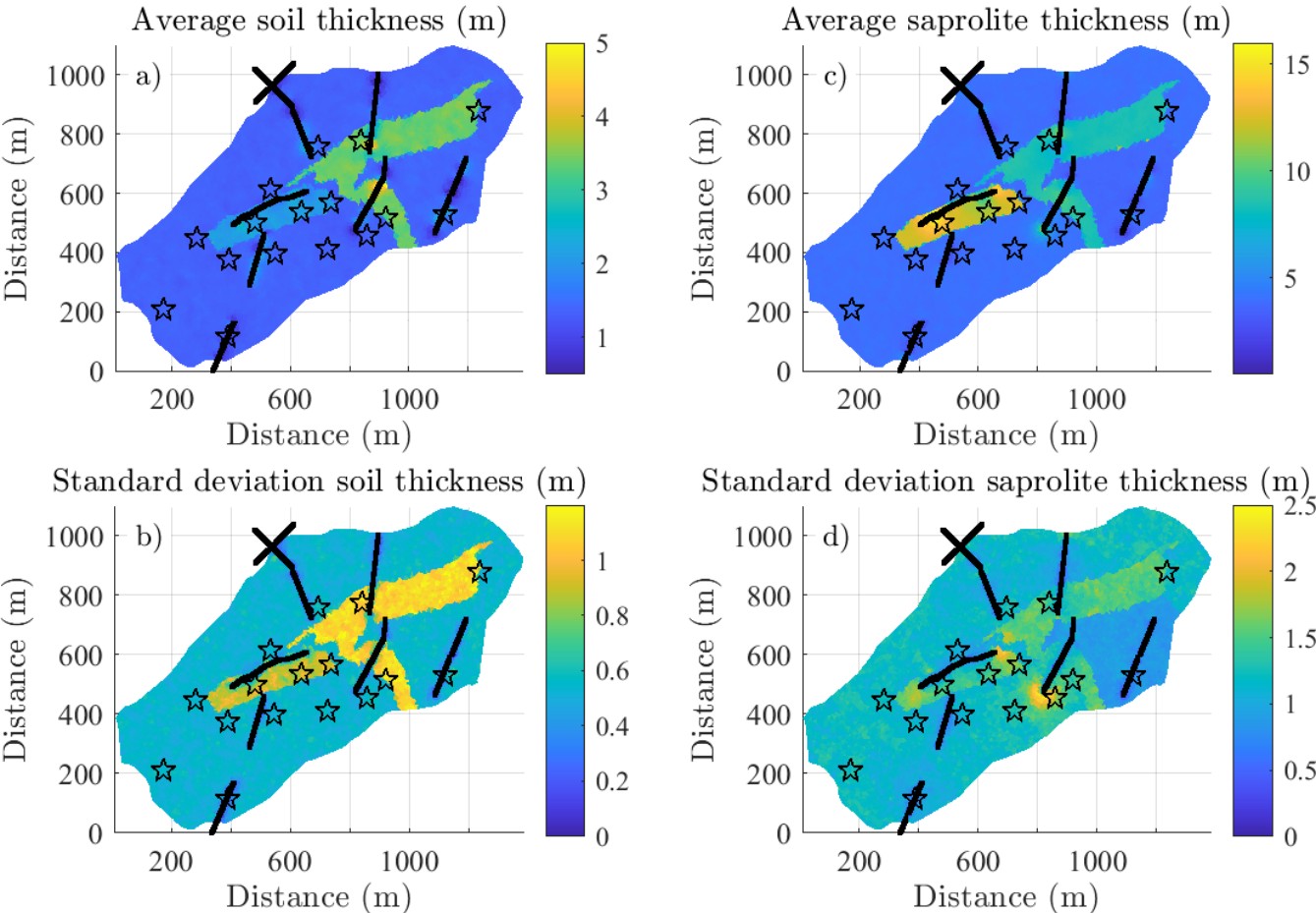

**Figure 9: Statistical characteristics of the lower boundary of the soil (a, b) and the saprolite (c, d). The averages (a, c) and the standard deviations (b, d) are estimated from the generation of 250 geostatical models following a Gaussian truncated power value geostatical model. The black dots represent the locations of the SRT profiles; the black stars correspond to the MRS station locations.**

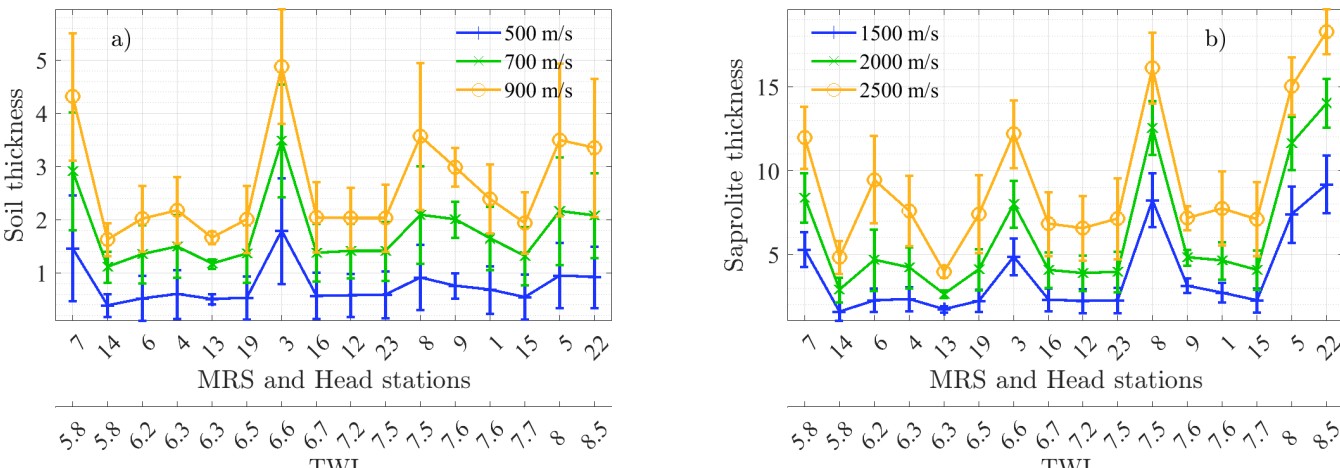

**Figure 10: Variation of the soil and saprolite thicknesses below each piezometric and MRS station. The thicknesses plotted represent the average estimated from the 250 generated fields, and the error bars correspond to the thicknesses' standard deviations. Stations are ordered with a crescent TWI.**

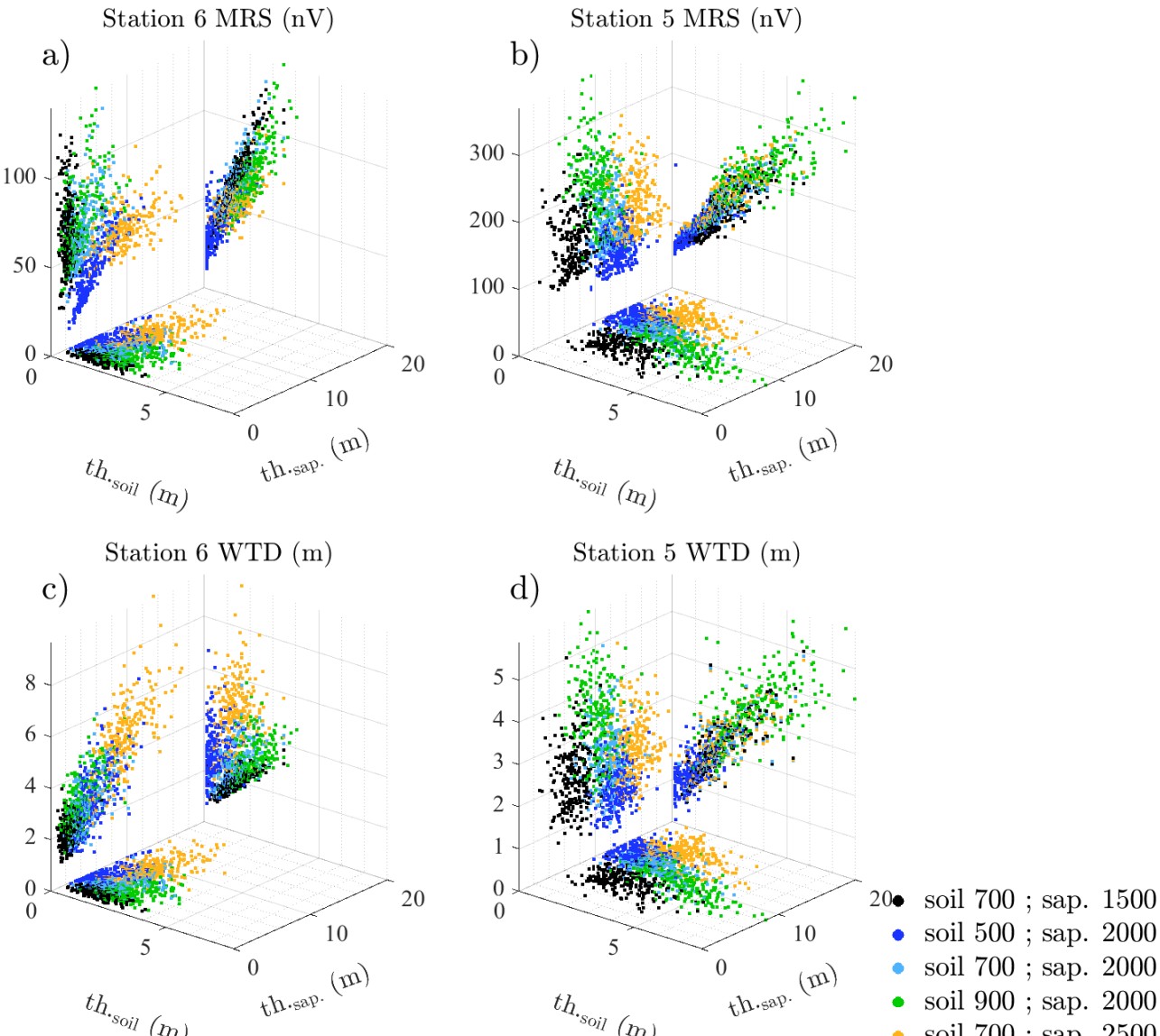

**Figure 11: Distribution of the MRS and WTD values as a function of the soil and saprolite thicknesses (labeled th.soil and th.saprolite) below measurement stations 6 and 5. Data are estimated the 19th of April 2013 for different velocity thresholds and the fixed set of parameter B (see Table 3).**

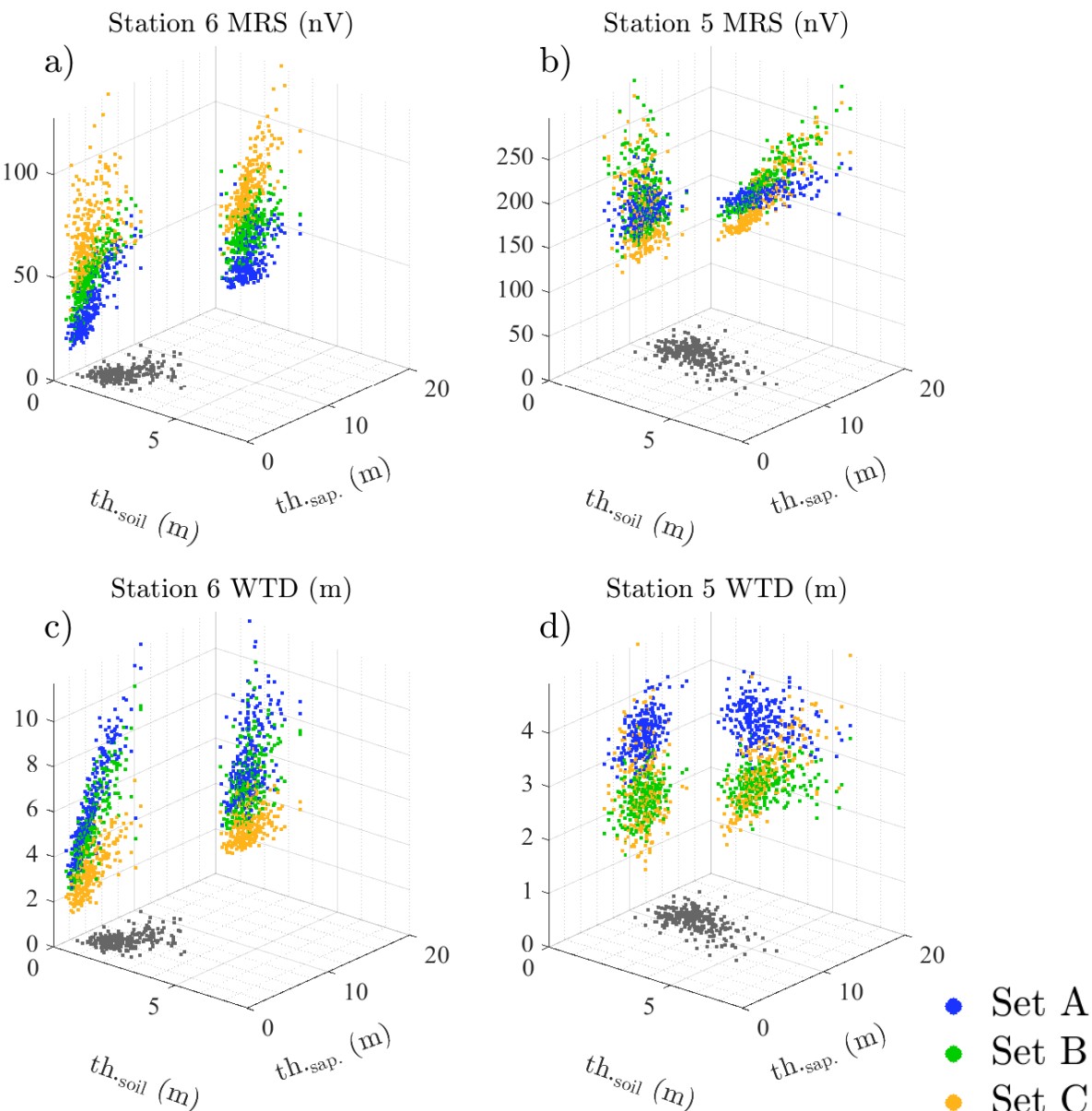

**Figure 12: Distribution of the MRS and WTD values as a function of the soil and saprolite thicknesses (labeled th.soil and th.saprolite) below measurement stations 6 and 5. Data are estimated the 19th of April 2013 for different sets of parameters, as described in Table 3, and fixed velocity thresholds of 700 m/s for the soil and 2000 m/s for the saprolite.**

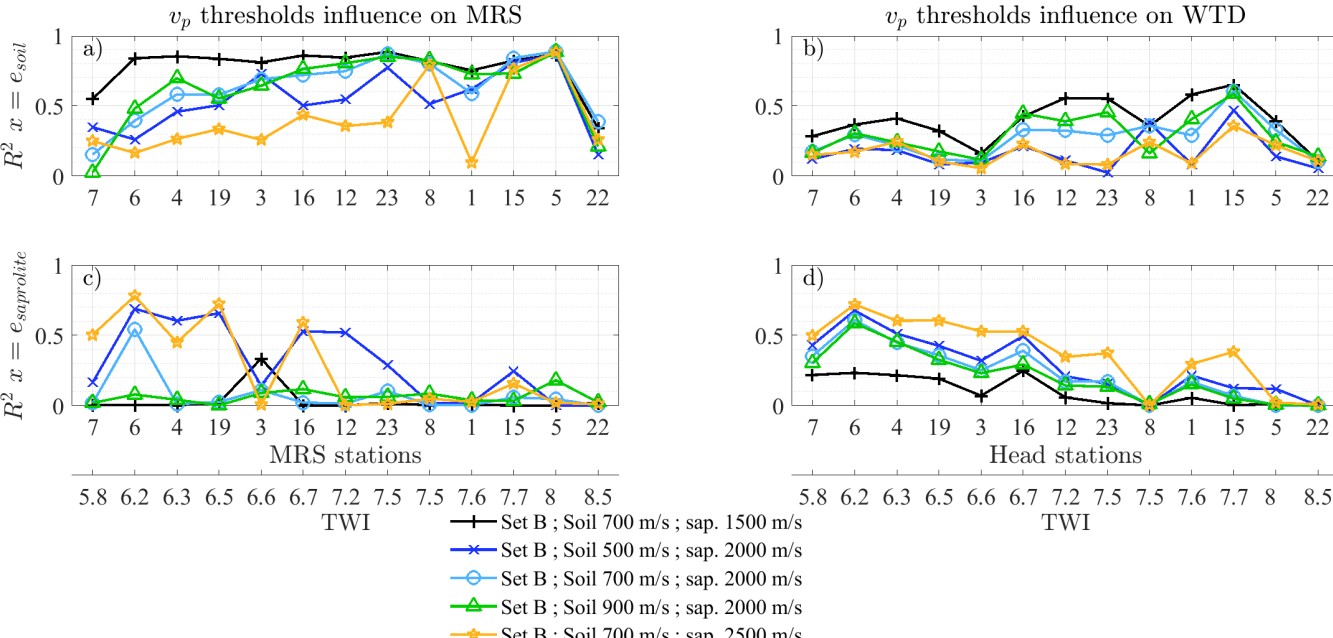

Figure 13: $R^2$ values of linear fits computed on the MRS and WTD signals estimated the 19th of April 2013 as a function of the thickness of the soil (a, b) and saprolite (c, d) below each station for different velocity thresholds and for the set B (see Table 3). Stations 9, 13 and 14 located close to the acquired seismic profiles are excluded from the analysis. Stations are ordered with a crescent TWI.

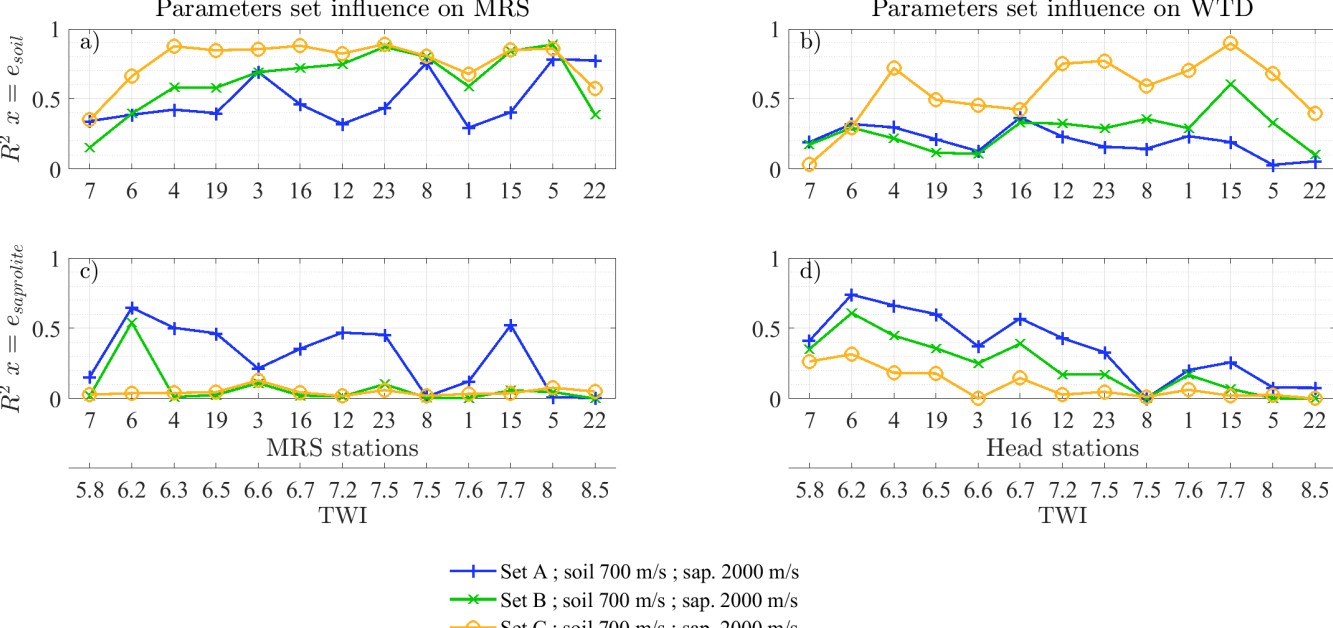

**Figure 14:** $R^2$ values of linear fits computed on the MRS and WTD signals estimated the 19th of April 2013 as a function of the thickness of the soil (a, b) and saprolite (c, d) below each station for different sets of parameters and fixed velocity thresholds of 700 m/s for the soil and 2000 m/s for the saprolite. Stations 9, 13 and 14 located close to the acquired seismic profiles are excluded from the analysis. Stations are ordered with a crescent TWI. The sets A, B and C correspond to the parameters' sets described in Table 3.

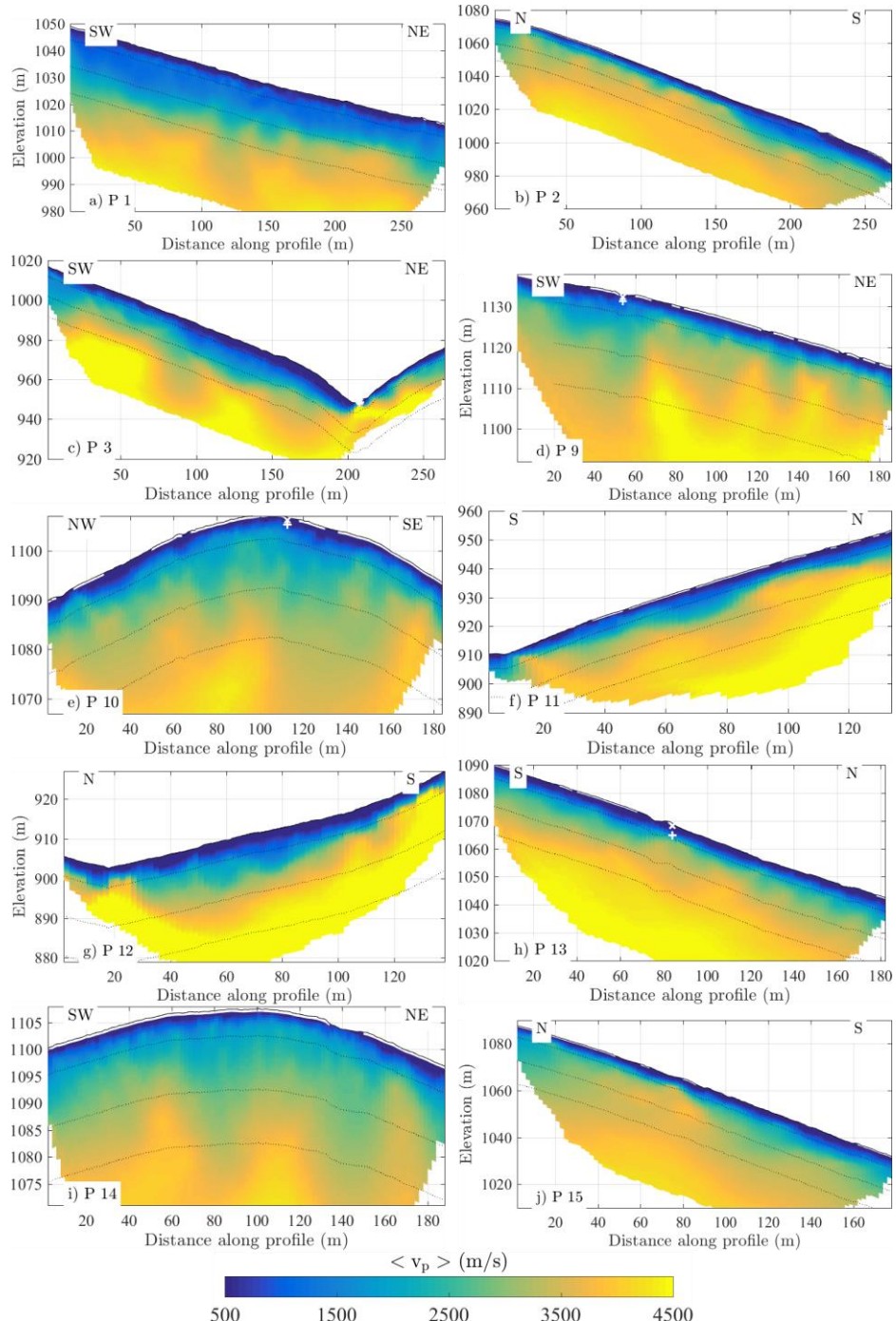

Figure A1: Average seismic velocity of the whole SRT profiles acquired on the Strengbach. The dotted lines correspond to the surface elevation minus 5, 15 and 25 m. The white cross (plus) indicates the depth at which the bottom interface of the soil (saprolite) was estimated during the drilling of a nearby piezometer or borehole.