# Peer review of "Impacts of hydrofacies geometry designed from seismic refraction tomography on estimated hydrogeophysical variables"

_EGUsphere, 2023_

## Author Response (AR1)

Dear Editor,

Thank you for your feedback and your request for minor revisions. In the new version of the manuscript we modified the text as requested by reviewers and described below. We also modified most of the figures so they are accessible to persons with colour vision deficiencies.

Best regards,

Nolwenn Lesparre on behalf of the authors.
* * *
**Jacopo Boaga comments:** *The work is very well presented and of interest for HESS. The statistical approach is robust and the findings of big impact for the hydrological studies of mountain environment. The overall process and correlations of SRT with MRS and other hydrological measurements are well developed, and the case studies deserves publication. The main criticism I have is about the raw input of the seismic data. Authors present just a concise description of the field dataset collection, without showing seismograms or processing phase of the SRT (only some in supplementary material). Authors assert they collect up to 144 channels surveys with 24 channels seismograph (roll?), with 2m spacing (total length up to 286 m, see fig.5), adopting a not clear 8-10 m offset. They used a weak 5kg sledge hammer. The acquisition scheme is not better clarified (roll? Sources? Stacking? Source locations?). By our experience in SRT in mountain slopes, it seems very ambitious to pick first arrivals with such a source over 90-100 m distance. This obviously implies the errors of picking, and then of the inverted section. Authors should provide more information about the raw data collected, presenting clear picked seismograms to prove the timing errors adopted (in paper tab and figures, not in the supplementary materials).*

**Answer:** We warmly thank the reviewer for his positive evaluation of our work. Following your recommendation, we added details about seismic acquisition in the main text. We also moved Table S1 to the main manuscript and added a new figure (called 2 bellow and in the new manuscript) with examples of picked seismograms. The inverted velocity models were inserted in a new appendix.

**Changes applied in the manuscript (changes in bold):**
(new manuscript, line 147) Ten SRT profiles, covering a total length of 2 km, were acquired in June 2018 and August 2019. Their locations were chosen to cover specific areas of the catchment, such as the valley bottom, the crests, the region upstream of the creek spring and both hillsides (Fig. 1). The surveys were designed to explore how the underground part of the CZ evolves in these different regions, which were previously distinguished by a joint analysis of pedological and MRS data collected across the catchment (Boucher et al., 2015; Lesparre et al., 2020a). **Seismic data were collected using up to 6 24-channel seismic recorders (Geometrics) and 14-Hz vertical-component geophones spaced with 2 m. For each profile, we used either 72, 96 or 144 geophones, for total lengths up to 142 m, 190 m and 286 m, respectively (Table 1). The source signal was generated with 4 stacks of a 5 kg sledgehammer blow on a metal plate, with shots**

**located every other 5 or 6 geophones, starting at first and ending at last geophone.**

**Table 1: Acquisition Parameters of the Seismic Lines**

| Line number | 1 | 2 | 3 | 9 | 10 | 11 | 12 | 13 | 14 | 15 |
|---|---|---|---|---|---|---|---|---|---|---|
| Number of traces | 144 | 144 | 144 | 96 | 96 | 72 | 72 | 96 | 96 | 96 |
| Trace spacing (m) | 2 | 2 | 2 | 2 | 2 | 2 | 2 | 2 | 2 | 2 |
| Line length (m) | 286 | 286 | 286 | 190 | 190 | 142 | 142 | 190 | 190 | 190 |
| Number of shots | 30 | 30 | 30 | 25 | 25 | 19 | 19 | 25 | 25 | 25 |
| Shot spacing (m) | 10 | 10 | 10 | 8 | 8 | 8 | 8 | 8 | 8 | 8 |
| Recording time (s) | 0.75 | 0.75 | 0.75 | 0.8 | 0.8 | 0.8 | 0.8 | 0.8 | 0.8 | 0.8 |
| Sampling time (ms) | 0.125 | 0.125 | 0.125 | 0.125 | 0.125 | 0.125 | 0.125 | 0.125 | 0.125 | 0.125 |
| Time delay (s) | -0.1 | -0.1 | -0.1 | -0.05 | -0.05 | -0.05 | -0.05 | -0.05 | -0.05 | -0.05 |

**First arrival times were picked manually on each shot gather. Signal-to-noise ratio varies significantly for each profile, but is mostly high enough to confidently identify first breaks up to 100-150 m distance from the source (Figure 2). This is more than enough to characterize the granite weathered zone anticipated to extend down to 10-15 m at most in such mountainous temperate catchment. The observed travel times were associated with a 5% picking error,** then used to build the subsurface P-wave velocity structure (vp) by solving an inverse problem with the pyGIMLi refraction tomography inversion module (Rücker et al., 2017). In pyGIMLi, the inversion domain corresponds to a triangular mesh with cells of constant velocity through which rays are traced using a shortest-path algorithm (Dijkstra, 1959; Moser, 1991). The velocity in each mesh cell is estimated using a generalized Gauss-Newton inversion framework. The inversion is iterative and starts with an initial model consisting of a velocity field that increases linearly with depth from [250 - 750] m/s at surface to [2000 – 5000] m/s in depth (Table S2). The velocity field is then smoothly updated at each iteration in order to reach the closest match between predicted and observed travel times. Inversions were performed with 144 combinations of starting models and regularization parameters (Table S2) in order to explore the possible solutions and estimate the

uncertainty of the velocity distribution along each profile (Pasquet et al., 2016).

[Figure]

Figure 2 Examples of shots along (a) Line 15 (96 geophones) and (b) Line 1 (144 geophones).

**Anonymous referee #2 comments:** *The authors propose a methodology to infer patterns of the subsurface critical zone at the catchment scale from seismic refraction data for hydrological modelling. The overall study appears to be a very good physically based distributed hydrological model applied to a mountainous catchment. As such, the study is highly relevant and fits within the scope of HESS.*

Thanks a lot for your appreciation of our work.

*The manuscript is well structured and well written. Nevertheless, I have some concerns regarding the thickness definition. In the NIHM model, transmissivity (\bar T) is obtained by integrating the hydraulic conductivity between $z_b$ and $z_w$ in the saturated zone and between $z_w$ and $z_s$ in the unsaturated zone - where $z_w$ is the hydraulic head with respect to the bottom $z_b$. The water content (\bar ϑ) is determined by integrating ϑ in the unsaturated zone, while the storativity (\bar S) is determined by integrating S between $z_b$ and $z_w$. Thus, by definition, \bar T, \bar ϑ and \bar S also depend on the state variable, i.e. the hydraulic head.*

Yes, we agree. $z_w$ is the water table elevation with respect to a reference which could be different from $z_b$

*On the other hand, on page 16 (lines 425-430), the authors state: "The equations defining the groundwater flows show that key hydraulic variables such as the transmissivity \bar T and the water content \bar ϑ correspond to the integration over the porous media thickness of the hydraulic parameters K(h) and \theta(h), respectively as stated in (6). Thus, to solve the inverse problem seeking the hydrological model parameters, misestimating the thickness of the hydrological model underground compartments would inherently lead to a wrong assessment of the hydraulic parameters."*

*In my opinion, the term "thickness" is misleading here. It is not clear whether the author is referring to saprolite or soil thickness, or saturated or unsaturated thickness. In fact, if the former, it is only necessary to rewrite the text, but if they mean the saturated or unsaturated thickness, the dependence of hydraulic head and MRS on both hydraulic parameters and saturated thickness is not new. Please clarify this aspect in the manuscript before proceeding.*

Thanks for your comment. The sentence needed to be clarified and we reformulated it by (line 430)**: "**Thus, to solve the inverse problem seeking the hydrological model parameters, misestimating **the soil and saprolite thicknesses** of the hydrological model would inherently lead to a wrong assessment of the hydraulic parameters**"**